



# Idealized Simulations of Mei-yu Rainfall in Taiwan under Uniform Southwesterly Flow using A Cloud-Resolving Model

Chung-Chieh Wang 1, Pi-Yu Chuang 1*, Shi-Ting Chen 1, Dong-In Lee 2, and Kazuhisa Tsuboki 3

1 Department of Earth Sciences, National Taiwan Normal University, Taipei, Taiwan
5   2 Department of Environmental Atmospheric Sciences, Pukyong National University, Busan, South Korea
Institute for Space-Earth Environmental Research, Nagoya University, Nagoya, Japan

*Corresponding to*: Pi-Yu Chuang (giselle780507@hotmail.com)

**Abstract.** In this study, idealized cloud-resolving simulations are performed for horizontally uniform and steady southwesterly flow at fixed direction/speed combinations to investigate rainfall characteristics and the role of the complex
topography in Taiwan during the Mei-yu season, without the influence of a front or other disturbances. Eight directions (180° to 285°, every 15°) and eight speeds (5 to 22.5 m s$^{-1}$, every 2.5 m s$^{-1}$) are considered, and near-surface relative humidity is also altered (from 55-100%) in a subset of these tests to further examine the effects of moisture content, yielding a total 109 experiments each having a integration length of 50 h. Three rainfall regimes that correspond to different ranges of the wet Froude number ($F_{rw}$) are identified from the idealized simulations (with a grid size of 2 km). The low-$F_{rw}$ regime
($F_{rw} \leq$ ~0.3) where the island circulation from thermodynamic effects is the main driver of rainfall in local afternoon during daytime. The lower the wind speed and $F_{rw}$, the more widespread and amount of rainfall. On the other hand, the high-$F_{rw}$ regime ($F_{rw} \geq$ ~0.4) occurs when the flow at least 12.5 m s$^{-1}$ impinges on Taiwan terrain at a large angle to favor the flow-over scenario. Thus, topographic rainfall production becomes dominant through mechanical uplift of unstable air. In this second scenario, the faster and wetter the flow, the heavier the rainfall on the windward slopes, with maximum amounts
occurring in directions typically from 240°-255°. Between the two regimes above, a third and mixed regime also exists. The idealized results are discussed for their applicability to the real atmosphere.

## 1 Introduction

The Mei-yu season in East Asia is a unique weather and climate phenomenon during the transition from the winter to summer monsoon, and it typically occurs from mid-May to mid-June in Taiwan (Chen, 1983, 2004; Ding, 1992). During this
rainy period that both provides vital water resources and at times brings heavy rainfalls and related hazards to the island, many mechanisms can lead to rainfall in Taiwan. The most obvious feature is the repeated passages of the Mei-yu front (e.g., Kuo and Chen, 1990), where the warm and moist tropical air mass encounters the colder and drier air from the north, and thus provides low-level convergence and frontal uplift to produce rainfall. The front not only brings an unstable environment to the region, at times it can accompany organized mesoscale convective systems (MCSs) such as intense rainbands to cause





heavy to extreme rainfall in Taiwan (e.g., Wang et al., 2016, 2021; Lupo et al., 2020). Furthermore, when a Mei-yu front
      approaches Taiwan, the prefrontal southwesterly flow often intensifies to form low-level jets (LLJs) in response to the
      enhanced horizontal pressure gradient (e.g., Chen and Chen, 1995), and the flow is subsequently uplifted by the steep and
      complex mesoscale terrain of the island (e.g., Lin, 1993; Jou et al., 2011). This is another common scenario for rainfall
      production in the mountain interior of Taiwan prior to the arrival of the surface front. In the past, many studies have
examined the roles of the front (e.g., Chen, 1993; Cho and Chen, 1995; Chen et al., 2008), the LLJ (e.g., Jou and Deng, 1992;
      Chen et al., 2005; Wang et al., 2014a), the topography of Taiwan (e.g., Lin, 1993; Wang et al., 2005), and the interactions
      among them (e.g., Lin et al., 2001; Xu et al., 2012; Tu et al., 2014; Wang et al., 2014b).

      In addition to forced uplifting, the steep topography of Taiwan also has another dynamical effect in terrain blocking (e.g.,
      Yeh and Chen, 2002, 2003). As airflow encounters an obstacle such as the topography of Taiwan, its overall response and
behavior are controlled by the Froude number ($F_r$), defined as $F_r = U/Nh_0$ (e.g., Pierrehumbert, 1984; Blumen, 1990; Baines,
      1995). Here, $U$ is the wind speed normal to the terrain, $h_0$ is the mountain height, and $N$ is the Brunt–Väisälä frequency and
      $N^2 = (g/\theta)(d\theta/dz)$, where $\theta$ is the potential temperature. In the low-$F_r$ regime ($F_r \leq 1$), the flow tends to be blocked and flow
      around the obstacle (e.g., Forbes et al., 1987; Bell and Bosart, 1988), and flow deflection occurs with the formation of
      ridge/trough on the windward/lee side (Smith, 1982; Banta, 1990; Overland and Bond, 1995). On the contrary, in high-$F_r$
regime when $F_r > 1$, the flow has enough momentum to climb over the terrain, and orographic precipitation can often be
      resulted (e.g., Manins and Sawford, 1982; Smolarkiewicz et al., 1988; Rotunno and Ferretti, 2003). In a Mei-yu case study,
      Wang et al. (2005) found that depending on the $F_r$, the blocking effect of Taiwan can shift the low-level convergence zone
      due to flow deceleration and deflection farther upstream (with higher $F_r$), thus causing rainfall over the plain area instead of
      near the mountains.

Besides the common ingredients of the Mei-yu front, southwesterly flow (LLJ), and the topography, other disturbances and
      mechanisms can also lead to rainfall in Taiwan. The island circulation that constitutes both the land-sea breeze and
      mountain-valley breeze (upslope-downslope wind) can develop under weak synoptic conditions with pronounced diurnal
      signals (Chen et al., 1999; Kerns et al., 2010). During daytime, sea breeze and upslope winds generate near-surface
      convergence and rainfall over the island, whereas offshore flow with divergence occurs at nights (e.g., Sha et al., 1991;
Johnson, 2011). Using the data collected during the South-West Monsoon Experiment (SoWMEX, Jou et al., 2011), Ruppert
      et al. (2013) found that the diurnal cycle in Taiwan is more pronounced during the undisturbed periods (without the front) in
      the Mei-yu season, and it is on average weaker but still exists during the disturbed periods (with the presence of the front).
      Thus, there are also thermodynamic effects of the topography, particularly under weak synoptic conditions.

      Other features and disturbances that also play various roles to produce or affect rainfall include disturbances along/near the
front (e.g., Chen, 1992; Chen et al., 2008; Lai et al., 2011; Wang et al., 2014b) and leeside mesolow and vortex (e.g., Sun
      and Chern, 1993, 1994; Wang et al., 2002, 2003). Preexisting disturbances embedded in the airflow (e.g., Davis and Lee,
      2012; Wang et al., 2018), often at the leading edge of stronger wind surges with convergence near the surface (Wang et al.,
      2014a) and gravity waves/density currents (e.g., Kingsmill, 1995; Fovell, 2005; Wang et al., 2011) are other possibilities. At





the storm scale, earlier convection (e.g., Nicholls et al., 1991; Walser et al., 2004; Wang et al., 2011; Xu et al., 2012) and even the interaction between vertical wind shear and updraft of mature cells are known to affect convective evolution and thus subsequent rainfall (e.g., Wang et al., 2016).

Most of the above studies were on real events, through either case studies, composites of similar cases after classification, or model simulations and sensitivity tests. Because each of the many influencing factors play a different role in different events, i.e., they are not controlled, it is very difficult to isolate the contribution from a single factor, or a small number of selected factors, and impossible to generalize the results for other events. For such a purpose, it is more effective to perform idealized simulations using numerical models, where unwanted features can be excluded and those included in the model can be properly controlled. Thus, idealized simulations are the approach adopted in this study.

Several idealized numerical studies were performed in the past, including those of Chu and Lin (2000) and Chen and Lin (2005b) in a two-dimensional (2D) framework, Chen and Lin (2005a) and Miglietta and Rotunno (2009) in three-dimensional (3D) space, and Sever and Lin (2017) in both. These studies investigated the effects of $F_r$ (i.e., wind speed) and the amount of Convective Available Potential Energy (CAPE) on conditionally unstable flow over a mountain ridge (mostly 2 km in height). They mainly identified several different regimes: flow with an upstream-propagating precipitation system with small $F_r$ ($\leq 0.5$) and large CAPE ($\geq 2000$ J kg$^{-1}$), stationary precipitation over the mountain with intermediate $F_r$ ($\leq 0.7$) and a wide range of CAPE, downstream-propagating orographic convection with larger $F_r$ (up to about 1.2), and flow over the terrain with stratiform precipitation typically with even larger $Fr$ (Chen and Lin, 2005b). While these studies cover the high-$F_r$ (flow-over) regime with strong winds (with $U \geq 36$ m s$^{-1}$), an idealized bell-shaped topography is used and effects of different wind directions are not investigated even in the cases of 3D simulations (which also have a limited dimension of only 10-20 km in the direction parallel to the terrain). In addition, the thermodynamic effects of the topography from radiation and the Coriolis effect associated with the earth's rotation are also turned off as controlled parameters in these experiments. Thus, while these studies help us gain better understanding on how conditionally unstable flow would respond when encountering a mountain, their results nevertheless are highly idealized and simplified.

Located between the Pacific Ocean and Eurasia continent, in the central area of East Asian monsoon (Fig. 1), Taiwan has a steep and complex topography as mentioned (also Figs. 2b,c). The long-term climatology (1981-2010) reveals abundant Mei-yu rainfall in the two-month period of May-June, with three maxima: two on the windward side of the Central Mountain Range (CMR) in southern and central Taiwan, respectively, and the third, less distinct center in northern Taiwan, roughly along the northern slope of the Snow Mountain Range (SMR). It is clear that the topography of Taiwan exerts strong control on the overall Mei-yu rainfall amount and distribution (also Kuo and Chen, 1990; Lin, 1993; Yeh and Chen, 1998; Chi, 2006) with significant diurnal variations (Chen et al., 1999; Kerns et al., 2010; Ruppert et al., 2013) as mentioned. Thus, certain aspects cannot be fully explored using idealized bell-shaped topography and without diurnal effects. For idealized simulation results to be more applicable to Taiwan, real topography and thermodynamical effects are both needed. This provides basic motivation for the present study, which has an objective to investigate the rainfall response under idealized southwesterly flow encountering the real topography of Taiwan that exerts both dynamical and thermodynamic effects. The prescribed





flow can have different direction and speed and moisture content that affects the CAPE and instability. Thus, the relative importance of dynamical and thermodynamic effects under different wind conditions can be assessed and the rainfall regime

in which one dominates the other (or vice versa) can be identified in a more generalized fashion.

The remainder of this paper is arranged as follows. The data and methodology, including the model and experimental design, are described in Section 2. In Section 3, our results of rainfall regimes under prescribed and uniform southwesterly flow are presented, and the influence of moisture and instability is discussed in Section 4. In Section 5, some of our idealized results are compared with real events to evaluate their applicability. Finally, the conclusions and summary are given in Section 6.

## 2 Data methodology

### 2.1 Sounding data and reference profile

In this study, a reference vertical profile of sounding and winds to represent the conditions upstream from Taiwan is first constructed. For this purpose, the sounding data at Dong-Sha Island and Research Vessel (RV) during the Southwest Monsoon Experiment in 2008 (SoWMEX, Jou et al., 2011) after quality control (Ciesielski et al., 2010) are used (cf. Fig. 1).

Screening is performed to exclude dates with synoptic disturbances (such as fronts and typhoons) near Taiwan, not in southwesterly flow regime (wind direction outside the range of 200°-270° or wind speed < 8 m s$^{-1}$ at 850 hPa), or with missing data. Eventually, soundings at 0000 UTC of seven dates are selected: 27-29 May and 1 June for Dong-Sha, and 28 May and 1 and 4 June for RV. Shown in Figs. 3a-d, the averaged thermodynamic, moisture, and wind profiles in the vertical from these data indicate a rather uniform south-southwesterly flow (8-13 m s$^{-1}$) that veers only slightly with height from the

lower to middle troposphere. The moisture content is high near the surface and the atmosphere is unstable (Figs. 3a,d), with a CAPE value of 2345 J kg$^{-1}$ and no Convective Inhibition (CIN) for a surface air parcel (at 1005.5 hPa).

Based on the mean sounding, the wind profile in the CTL are modified to give a uniform southwesterly flow of 10 m s$^{-1}$ at 240° from 950 to 500 hPa, and changes linearly to a prescribed profile at 300 hPa and above based on the observation (Figs. 3f,g). Below 950 hPa, the wind is set to change linearly downward from 950 hPa, to half the speed and 15° to the left at the

surface due to friction, also in close agreement with the observation. While the temperature ($T$) profile is unchanged, the prescribed moisture profile (Fig. 3h) with a relative humidity (RH) of 85% from the surface to 950 hPa and 40% at 500 hPa and above (changed also linearly in between) give a thermodynamic diagram as in Fig. 3e and raise the CAPE to 2803 J kg$^{-1}$. This value is comparable to those found in some previous studies (e.g., Wang et al., 2005). The above method used to construct the idealized (reference) wind profile is summarized in Table 1 (top half).

### 2.2 Idealized initial and boundary conditions

The prescribed and smoothed sounding profiles as described above (Figs. 3e-h; every 25 and 50 hPa below and above 500 hPa, respectively) are used to construct the 3D initial and boundary conditions (IC/BCs) of the control (CTL) experiment.





This reference sounding is assumed to be at 23.5°N, 120.5°E (near central Taiwan, cf. Fig. 1). From this point, the geostrophic wind relationship is used to determine the geopotential height (Φ) of a grid every 0.25° × 0.25° inside the

rectangular area of 16°-31°N, 110°-131°E at each pressure ($p$) level (at and above 950 hPa) as:

$$\mathbf{V}_g = -(1/f)\,(\partial\Phi/\partial n) \tag{1}$$

where $\mathbf{V}_g = (u\,\mathbf{i},\,v\,\mathbf{j})$ is the geostrophic wind vector, $f$ is the Coriolis parameter, and $n$ is the distance in normal direction (to the left) of the wind. Thus, on each $p$-level, $u$, $v$, $T$, and RH are all uniform, but Φ is not. Below 950 hPa down to the surface, the value of $\partial\Phi/\partial n$ at 950 hPa is used instead of its own level to include friction. As time-invariant conditions are provided

during the entire course of model simulation, the BCs are identical to the IC in the CTL (as well as in each of all other experiments). In addition to the meteorological fields, digital terrain data on a (1/120)° grid and the time-mean sea surface temperature (SST) analyzed by the National Oceanic and Atmospheric Administration (NOAA) using optimal interpolation (Reynolds et al., 2002) are also provided at the lower boundary (Table 2). These conditions are identical in all model runs.

### 2.3 The cloud-resolving model

In this study, the Cloud-Resolving Storm Simulator (CReSS) version 2.3 (Tsuboki and Sakakibara, 2002, 2007) is used for all model experiments. The CReSS model employs a non-hydrostatic and compressible equation set and a terrain-following vertical coordinate, and is designed to simulate clouds at high resolution. Thus, all clouds are treated explicitly in CReSS using a 1.5-moment bulk cold-rain microphysics scheme, which is based on Lin et al. (1983), Cotton et al. (1986), Ikawa and Saito (1991), and Murakami et al. (1990, 1994) and includes a total of six water species (vapor, cloud water, cloud ice, rain,

snow, and graupel). A warm-rain scheme that has no ice phase is also available but not used here. As given in Table 2, the parameterized processes at the sub-grid scale include turbulent mixing in the planetary boundary layer (Deardorff, 1980; Louis et al., 1982), surface shortwave/longwave radiation, and surface momentum and heat fluxes (Kondo, 1976; Sagami et al., 1989). A substrate model is also included (Tsuboki and Sakakibara, 2007). The model is open for research, and its further details can be found online (http://www.rain.hyarc.nagoya-u.ac.jp/~tsuboki/ cress_html/index_cress_eng.html) or in some

earlier studies (e.g., Tsuboki, 2008; Wang et al., 2014a,b, 2016).

### 2.4 Experimental design

In CTL, where the southwesterly winds are from 240° at 10 m s⁻¹ over the depth of 950-500 hPa, the idealized IC/BCs are provided to the CReSS model as described earlier. At a horizontal grid size of 2 km, the CReSS model then simulates the atmospheric evolution inside a domain surrounding Taiwan (roughly over 18°-28.2°N, 112.5°-125.8°E; Fig. 1 and Table 2).

To investigate the change in flow regime and rainfall, eight different wind directions and eight different wind speeds are tested for 950-500 hPa: every 15° from 180° to 285° and every 2.5 m s⁻¹ from 5 to 22.5 m s⁻¹, yielding 64 experiments for this purpose. The IC/BCs are constructed individually for each experiment with the same $T$ and RH profiles (as in Figs. 3e,h). For each experiment, the wind is fixed at the same direction and speed over 950-500 hPa as prescribed (Table 1, top half),



and then varies linearly to 300 hPa, where the same profile further up (as in Figs. 3f,g) is used for all runs. Similarly, the
wind gradually reduces in speed and turns to the left below 950 hPa, as described earlier.

To further examine the effect of near-surface moisture content, nine experiments from the above tests, with wind directions of 210°, 240°, and 270° and wind speeds of 10, 15, and 20 m s$^{-1}$, are selected for this purpose. These combinations of direction/speed are chosen to both include the CTL and provide a wide-enough range of variations for comparison. At the lowest levels from surface to 950 hPa, the RH is changed from 85% to other values every 7.5%, from as dry as 55% to as
moist as 100% (except for 62.5%). Above 950 hPa, RH values are reduced linearly to 40% at 500 hPa, and the same RH profile is used above that, as shown for some examples in Fig. 3h (and Table 1, bottom half). These tests thus include an additional 45 experiments (9 wind combinations × 5 different RH levels besides 85%).

Starting from 2200 UTC, each simulation is run for a length of 50 h, which allows for a 2-h spin-up period (2200-2400 UTC of day 0) plus two full-day cycles (days 1 and 2, cf. Table 2). Essentially, all experiments produce two similar diurnal cycles
during 2-50 h, thus their daily averages (over days 1-2) will be the main subject for discussion.

**2.5 Result analysis and comparison**

To identify the flow regime associated with each combination of wind direction/speed, the moist Froude number ($F_{rw}$), which uses virtual potential temperature ($\theta_v$) and $N_w^2 = (g/\theta_v)(d\theta_v/dz)$ instead, are computed for each case, as in Chen and Lin (2005). Compared to the dry $F_r$, $F_{rw}$ takes into account the effect of moisture on density, since the atmospheric
environment near Taiwan is often very moist in the Mei-yu season (cf. Fig. 3). Considering the highest topography in the north-south profile (Fig. 2c), a value of 2.5 km is used for $h_0$. The $N_w$ is also computed for the lowest 2.5 km, while the prescribed wind (same over 950-500 hPa) is used to obtain $U$. Finally, some real cases of southwesterly flow and daily rainfall are chosen to compare with our idealized results in Section 5. For this purpose, the National Centers for Environmental Prediction (NCEP) Global Forecast System (GFS) final analyses (Kalnay et al., 1990; Kleist et al., 2009) at
850 hPa, inside a 2° × 2° box near Dong-Sha (cf. Fig. 1), are used to identify and classify the southwesterly flow. Only the 0000-UTC data on each day in the Mei-yu season in 2012-2014 are used.

**3 Results of prescribed and uniform southwesterly flow**

**3.1 Control experiment**

The results of the CTL is first examined in this section, in order to characterize the behavior of the model under the idealized
conditions and confirm that it behaviors as designed. In Fig. 4, horizontal wind and pressure fields at the surface at selected times every 3-5 h are presented, and those at the model level of 1481 m (close to 850 hPa) are shown in Fig. 5 at longer intervals. At the initial time ($t = 0$ h), it can be confirmed that the winds are uniform and parallel to the isobars at 1481 m (Fig. 5a), but at 15° across the isobars at the sea level (Fig. 4a). However, within a few hours into the integration (Figs. 4b,c



and 5b), the flow upstream quickly decelerates and separates into two branches to flow around the topography due to the

blocking effect once it encounters the obstacle, as expected since the $F_{rw}$ is only 0.28 in CTL (cf. Table 3). At the two ends of Taiwan, the flow converges and accelerates to form barrier jets near the northwestern coast and off southeastern Taiwan (Figs. 4c-f and 5c,d), in agreement with many earlier studies (e.g., Li and Chen, 1998; Yeh and Chen, 2002, 2003; Wang et al., 2016). This low-level flow and pressure pattern remain rather steady through time after model spin-up in CTL (Figs. 4 and 5), except for eastern Taiwan where a leeside low and vortices develop and evolve (e.g., Wang and Chen, 2002, 2003).

As rather transient phenomena, the formation of vortex pairs and vortex shedding are also clearly visible in Fig. 5. Overall, the model behaves as designed in the CTL and the results are consistent with many previous studies (also Sun and Chern, 1993, 1994).

The rainfall in CTL occurs mostly over Taiwan during 0300-1100 UTC (or 1100-1900 LST, Figs. 4c,d and i,j), mainly in local afternoon, also in agreement with the climatology and many earlier studies (e.g., Chen et al., 1999; Kerns et al., 2010;

Ruppert et al., 2013) but in contrast to previous idealized results without diurnal effects (e.g., Chu and Lin, 2000; Chen and Lin, 2005a,b; Sever and Lin, 2017). Over the ocean, very little rain is produced in CTL, with rather uniform flow upstream. This suggests the sole role of the terrain in triggering convection to lead to rainfall through either its dynamical or thermodynamic effects (or a combination of both), as designed in this study. The time series of hourly rainfall averaged over Taiwan also indicate two similar diurnal cycles in CTL (Fig. 6, red). However, compared to the observed rainfall cycle

compiled from the dates of the sounding data, the model seemingly produces too little rainfall. This is understandable, because the rainfall mechanisms in the model are only those associated with the topography of Taiwan by design, whereas the clouds and rain can also form by other mechanisms and move in from surrounding oceans in real events.

### 3.2 Rainfall regimes of uniform southwesterly flow

The (averaged) daily rainfall distributions in the 64 experiments of eight wind directions and eight wind speeds are presented

in Fig. 7, where the $F_{rw}$ is also given (and in Table 3). Nearly parallel to the topography of Taiwan (set to 16.7°-196.7°), the flow from 195° gives near-zero $F_{rw}$ values regardless of the speed (Table 3). Thus, the flow from 180° (southerly) and 210° (south-southwesterly) have slightly larger $F_{rw}$ values that are comparable to each other and also increase with wind speed (and up to about 0.25). As the flow direction becomes more westerly and perpendicular to the topography, the $F_{rw}$ further increases, to a maximum value of 0.91 at 22.5 m s$^{-1}$ from 285° (Table 3). Due to the high terrain of Taiwan ($h_0$ = 2.5 km),

the $F_{rw}$ never reaches unity. It is perhaps also worthwhile to note that, at a near-surface RH of 85% in these experiments, the $F_{rw}$ is only about 4% larger than the dry $F_r$, so their differences are rather small.

Based on the rainfall pattern and amount in Fig. 7, the results here are classified into three regimes: The low-$F_{rw}$ regime, the (relatively) high-$F_{rw}$ regime, and a mixed regime in between. In the low-$F_{rw}$ regime, the island circulation arising from the thermodynamic effects is the main driver to cause rainfall, over one or both sides of the mountain (by upslope winds) and

possibly also over the western plains (by see breeze, cf. Fig. 2b). This regime includes all conditions with lower wind speeds





of 5.0-7.5 m s$^{-1}$ and at a higher wind speed when the flow is at a small angle to the terrain, with a $F_{rw}$ about 0.3 at most (Table 3). In general, the amount and spatial coverage of the rainfall increase in this regime when the flow is weak and at a smaller angle (nearly parallel) to the topography (Fig. 7), and thus the conditions are favorable for the development of a stronger island circulation (e.g., Akaeda et al., 1995; Chen et al., 1999; Kerns et al., 2010).

The second mode of rainfall is with a relatively high $F_{rw}$ of roughly 0.4 and above, when the flow is at least 15 m s$^{-1}$ in speed and impinges on the terrain at a large angle of ≥225° (Table 3 and Fig. 7). Under such scenarios, significant rainfall occurs in the mountain interiors of central Taiwan (near the intersection of SMR and CMR) and southern Taiwan along the ridge of the CMR (Fig. 7, bottom half), producing a pattern not unlike the climatology (cf. Fig. 2a). The rainfall also increases with wind speed and reaches a peak amount when the wind direction is near 255°. Evidently, the dynamical effect of terrain uplift

is the dominant rain-producing mechanism in these high-wind conditions, as the rainfall becomes persistent with small diurnal variations (cf. the example of 20 m s$^{-1}$ from 240° in Fig. 6). At the highest speed of 22.5 m s$^{-1}$, such orographic rainfall can also take place when the flow is at a small angle (from 180° or 210°) with $F_{rw}$ below 0.3 (Fig. 7h), since the southernmost part of the CMR is not as high (cf. Fig. 2). Note, nevertheless, that when the flow is from 180° (210°), the eastern (western) slope of the CMR is the windward side and where the rainfall mostly occurs.

In between the above two rainfall regimes, there is a third, mixed regimes of both mechanisms and rainfall characteristics, as also labeled in Table 3 (cells with no color). Such a mixed regime occurs at high speed (≥17.5 m s$^{-1}$) but small angles with low $F_{rw}$ (about 0.15-0.25), or at medium speed (10-15 m s$^{-1}$) but larger angles with $F_{rw}$ around 0.3-0.4. Thus, the $F_{rw}$ values to separate the two major rainfall regimes are not the same, and smaller (greater) when the flow is at a smaller (larger) angle to the terrain. Also, the more perpendicular the flow is to the topography in the mixed regime, the rainfall tends to be less, although the differences are often relatively small. Presumably, this is because of less contribution from the island

circulation as well as a stronger blocking effect on the flow (even though the $F_{rw}$ increases).

**3.3 Rainfall regimes and their rainfall amounts**

In this subsection, the three rainfall regimes are further discussed more quantitatively, and with the information on the sub-region of rainfall maximum in Taiwan (cf. Fig. 2b). Table 4 gives the daily mean rainfall (spatially averaged) and peak

amounts over Taiwan, as well as the sub-region where the peak amount occurs in each of the 64 experiments.

When the wind speed does not exceed 12.5 m s$^{-1}$, the mean rainfall over Taiwan decreases with increasing wind speed (Table 4) across the entire spectrum of wind directions from 180° to 285°, and this covers mainly the low-$F_{rw}$ regime. Compared to other directions, the flow at 195° always produces the highest daily mean rainfall in Taiwan (under this regime), which can be up to 6.31 mm at 5 m s$^{-1}$ (Table 4). This is however only slightly higher than those values associated with other

wind directions, which are expectedly less relevant at such a low speed. When the wind speed goes slightly higher to ≥7.5 m s$^{-1}$, the differences between 195° (3.24-5.64 mm) and other directions become more evident (Table 4). Even at 15-17.5 m s$^{-1}$, the flow from 195° (low-$F_{rw}$ regime) still produces more rain than 180° and 210° (possibly in a different regime), although


its mean value further decreases. Thus, at low wind speeds, the most rainfall is produced by the flow from 195° at 5 m s⁻¹, with a maximum daily amount of 183 mm in central Taiwan. Similar conditions produce slightly less rainfall, with the peak

value (~150 mm) also often in central Taiwan (Table 4), likely linked to the higher mean elevation and more compact topography (i.e., closer proximity of sea breeze and upslope winds, cf. Fig. 2b). As the wind direction changes from southerly to westerly, the sea breeze and upslope winds on the western side become less able to develop (cf. Fig. 7), especially when the wind speed also increases higher, and the region of maximum rainfall shifts to northern or eastern Taiwan (Table 4). These two sub-regions tend to be better shielded by the high topography under westerly flow, and upslope

winds there are apparently less affected. Thus, even at the same flow speed, the rainfall amount and pattern also exhibit considerable sensitivity to wind direction. Similarly, at higher wind conditions of 10-17.5 m s⁻¹, the island circulation becomes more difficult to develop over the western part of the island (especially over the southern plains), the rainfall areas shift toward the northern and eastern sub-regions with less overall amount, even though the peak value can remain quite significant (around 100-180 mm). Here, it is also noted that the peak rainfall occurs in the elevation range of ≥1 km (over the

mountains) in its respective sub-region in all 64 runs Table 4 without any exception.

In the high-$F_{rw}$ regime where the convection triggered by mechanical uplift over the mountains is the major source of rainfall, the conditions are somewhat more straightforward and less complicated. From 12.5 to 20 m s⁻¹, the overall rainfall in Taiwan increases with the speed for flows coming from 210°-285° as expected, with only a few exceptions (Table 4), and generally maximizes at 255° as mentioned. Nevertheless, at 12.5 m s⁻¹, the flow from 255°-285° is at a large angle and close to normal

to the topography and thus possesses a relatively high $F_{rw}$ (around 0.5), but the mean rainfall in Taiwan (below 0.4 mm) is among the lowest in all experiments. This sensitivity to wind direction indicates that significant blocking by the terrain under such conditions can deflect the prevailing flow and shift the rainfall area further upstream (and offshore, cf. Fig. 7d), as shown by Wang et al. (2005), even though an increase in $F_{rw}$ (from lower wind speeds) favors the flow-over regime in theory. As a result, the small peak values (~25 mm) take place in the eastern sub-region (as for the wind-speed cases of 10 m s⁻¹).

Therefore, the blocking effect is another factor that can come into play and affect rainfall pattern. As the wind speed and $F_{rw}$ further increase (to ≥ 17.5 m s⁻¹), the mechanical uplift and flow-over regime become more dominant, and both the mean rainfall and its peak value increase rapidly with wind speed (Table 4), to maxima of 14.76 mm (at 255°) and 578 mm (at 240°) at the highest speed of 22.5 m s⁻¹, respectively. The locations are almost exclusively in the mountain interior of southern CMR. Similar heavy to extreme rainfall events with daily maximum in excess of 500 mm over the mountains are

also observed in the Mei-yu season (e.g., Wang et al., 2016). Finally, as suggested in Table 4, strong southern flow (at 180°) can also lead to significant rainfall in eastern or southern Taiwan, up to a mean value of >5.5 mm and a peak amount of over 300 mm.

In the mixed regime, the overall rainfall tends to be less with a peak value in eastern Taiwan, when the wind is around 10 m s⁻¹ and from 240°-285° (Tables 3 and 4), as mentioned earlier. With the flow at 15 m s⁻¹ from 225° and at 17.5 m s⁻¹ from

210°, the north sub-region located at the leeside receives the most rainfall, which peaks at 156 mm in the former case. For





the two cases with the flow from either 180° or 210° at 20 m s$^{-1}$, a transition rainfall pattern occurs (cf. Fig. 7g) from flow-around to flow-over regime, and the peak rainfall is produced in western and eastern sub-regions, respectively (Table 4). Thus, the peak rainfall area is typically at the leeside in the mixed regime, while the detailed rainfall distribution can be quite variable and rather complex (cf. Fig. 7).

To summarize the above results, the averaged daily rainfall in the three elevation ranges over Taiwan and the four sub-regions are plotted in Fig. 8 as examples, for three cases with southwesterly flow from 210° at 7.5 m s$^{-1}$ ($F_{rw} = 0.07$), 225° at 12.5 m s$^{-1}$ ($F_{rw} = 0.24$), and 255° at 20 m s$^{-1}$ ($F_{rw} = 0.69$), respectively. In the first case (Fig. 8a) where the island circulation controls rainfall production, considerable rainfall is received in northern, central, and eastern Taiwan, and in all three elevation ranges. At 12.5 m s$^{-1}$ from 225°, the second case has a $F_{rw}$ of 0.24 and is close to the transition, and the major

rainfall area is in northern Taiwan at the leeside (Fig. 8b). Finally, when the $F_{rw}$ is large, heavy rainfall occurs in the mountains over southern and central Taiwan through terrain uplifting (Fig. 8c).

## 4 Effects of near-surface moisture on rainfall

Under a uniform and fixed southwesterly flow, the amount of low-level moisture acts as another influencing factor although intuitively its primary role is to change the rainfall amount. Therefore, the facet of moisture content is investigated in this

section to complement the study thus far and make it more complete. As discussed in Section 2 and shown in Fig. 3h and Table 1 (bottom half), a total of 45 experiments are performed to change the near-surface RH inside the PBL from 85%, to a higher value of 92.5 or 100% and a lower one of 77.5, 77, or 55%, respectively. These five sets of runs are for fixed southwesterly flows at 10, 15, and 20 m s$^{-1}$ and from 210°, 240°, and 270° (nine runs in each set of fixed RH). As given in Table 5, the near-surface RH value affects the CAPE, which can be over 5500 J kg$^{-1}$ for RH = 100% and reduces to 464 J

kg$^{-1}$ for RH = 70% or even zero for RH = 55%. On the other hand, the impacts of RH on $F_{rw}$ are at most only about ±1% from those given in Table 3, and are therefore negligible. For this reason, the same $F_{rw}$ values as before are used for discussion.

### 4.1 Effects of moisture increase

The results of mean daily rainfall distributions in the four sets of different near-surface RH values from 100% to 70%

(excluding 85%) are presented in Fig. 9 and can be compared with the corresponding panels in Fig. 7. As the RH is altered, the changes in peak rainfall amount and its sub-region are listed in Table 6 and can be compared with Table 4. In cases where the RH and CAPE are increased, the rainfall amount and spatial coverage both increase as expected, particularly at higher wind speed (15-20 m s$^{-1}$) and near-surface RH reaches 100% (Figs. 8c,e,g and 9a,b and Table 5). From 210° at a smaller angle, a saturated condition promotes convection at the windward side of the CMR by strong flow, and shifts the

peak rainfall to southern Taiwan (Table 5, also all in mountain). At an angle more perpendicular to the terrain (240° and


270°), higher RH also increases the rainfall at the windward side, especially in central and southern Taiwan, and also over the nearly oceans in the upstream area (Figs. 9a,b). In some cases, east-west oriented rain belts are produced across Taiwan from the convection triggered upstream, including the plain areas. While the sub-region of peak rainfall remains at southern Taiwan in such a scenario (≥ 240° and 15-20 m s$^{-1}$), the peak daily rainfall amount can reach 749 mm for the case of RH =

92.5% and further to 994 mm for RH = 100% (both at 20 m s$^{-1}$ from 240°), respectively (Table 5). Thus, except for more rainfall, the increase in near-surface RH also plays a role to trigger convection more easily over the windward sides and upstream areas, and subsequently promote rainfall in those regions over and near Taiwan.

### 4.2 Effects of moisture decrease

Three sets of 3 × 3 experiments are also performed to test the response when the near-surface RH is reduced from 85%, to

77.5, 70, and 55%, respectively. In these tests, the changes in rainfall over Taiwan are quite straightforward, including a reduction in both amount and areal extend, without much difference in its general pattern (Figs. 8c,e,g and 9c,d and Table 5). Besides a reduced rainfall, some noticeable deviations include a shift of maximum rainfall sub-region from southern to eastern Taiwan under the flow of 210°/20 m s$^{-1}$, as the RH is lowered from 85% to 77.5% and less (Table 5). In this case of RH = 77%, the peak rainfall is located at the southernmost part of CMR, which is classified into the eastern sub-region by

our simple method. Nevertheless, this is resulted because the convection becomes more difficult to be triggered and thus less active at the windward side when the RH is reduced, thereby causing a shift in the sub-region of peak amount. Other differences are more subtle and often linked to slightly different responses of rainfall centers. Thus, the findings here are also in agreement with those in the previous sub-section with increased RH.

### 5 Comparison of idealized results to real events

Although the results of the present study are idealized model simulations with prescribed and uniform southerly to westerly flow (from 180° to 285°) and near-surface moisture content (from RH = 55% to 100%), it is perhaps worthwhile to explore how applicable these results are to the real atmosphere with actual topography. Therefore, such a discussion is provided in this section, by comparing our rainfall results to those observed during the Mei-yu season in Taiwan. While a wide spectrum in the combinations of wind direction, wind speed, and moisture amount are simulated in Sections 3 and 4, some conditions

in the spectrum (such as low moisture content) are not frequent and readily available in observation. Eventually, three sets of scenarios, with three cases in each, are selected below for comparison: flows with increasing speed from 210° in low-$F_{rw}$ regime ($F_{rw} \le 0.12$), flow with increasing angle from low-$F_{rw}$ to almost the mixed regime ($F_{rw} \le 0.26$), and, from low-$F_{rw}$ to high-$F_{rw}$ regime at 210° to 240° ($F_{rw} = 0.12$ to 0.56). It should be noted that, since the conditions are often more complicated in real events (where various disturbances exist, and the flow is neither horizontally uniform nor steady in time), the goal of

such comparisons is not so much in how closely the model results resemble the observation. Rather, it mainly focuses on





whether similar changes in rainfall pattern are found in both model result and observation, as a response to the changing flow conditions.

### 5.1 Flow from 210° in low-$F_{rw}$ regime

The comparison between model-simulated and observed daily rainfall in this low-$F_{rw}$ scenario is shown in Fig. 10 for the
southwesterly flow from 210° at three different wind speeds: 5, 7.5, and 12.5 m s$^{-1}$, respectively. The three corresponding dates in the observation are 26 May 2013, and 26 and 25 Jun 2012 following the order, so chosen as they also exhibited 850-hPa winds (in NCEP analyses) matching the specified conditions using the method described in Section 2.5.

As the flow direction remains at 210°, the $F_{rw}$ is proportional to wind speed but still very small at 0.05, 0.07, and 0.12, respectively, so the thermal effect and island circulation clearly dominate (Figs. 10a-c), similar to the undisturbed periods in
Ruppert et al. (2013). While the near-surface moisture differed slightly from the prescribed value, the observations (from 0000-2400 LST, Figs. 10d-f) also show similar characteristics without the maximum rainfall near the mountain ridge. As the wind speed increases from 5 to 12.5 m s$^{-1}$, the idealized model results indicate a reduction in overall rainfall in Taiwan, especially in southern and then central Taiwan, and thus a tendency for northern Taiwan (being more at the leeside) occupy a higher percentage of total rainfall (Figs. 10a-c). These tendencies are also discernable in the observations (Fig. 10d-f). For
more detailed comparison, the model produces some rainfall associated with the sea-breeze front over the plains in both central and southern Taiwan (about 50 km inland) when the flow is weak at only 5 m s$^{-1}$, and the southern one nearly diminishes at 7.5 m s$^{-1}$ and both disappear at 12.5 m s$^{-1}$. This tendency is also in general agreement with the observation and very encouraging. However, the model has too much convection along the eastern slope of the CMR, and at times not enough rainfall in the mountain interior of central Taiwan. As noted earlier, such discrepancies can be resulted from many
differences between idealized and real flows. In the real events, some rainfall occurrence (e.g., afternoon convection) may also be linked to different preconditioning of the local environment (e.g., Nicholls et al., 1991; Walser et al., 2004; Wang et al., 2011), i.e., what happened or did not happen on the previous day. Of course, such differences are not considered in the idealized framework.

### 5.2 Flow with an increasing angle to topography

The three cases in the second set are compared in Fig. 11, for the combinations of 12.5 m s$^{-1}$ from 195°, 10 m s$^{-1}$ from 225°, and 7.5 m s$^{-1}$ from 255°, following this order. The values of $F_{rw}$ are 0.01, 0.19, and 0.26, respectively, while corresponding dates in observation are 22 Jun 2012, 15 May and 9 Jun 2013. These conditions are still in the low-$F_{rw}$ regime, but they are closer to the mixed regime (especially the last case) and the increase in $F_{rw}$ mainly comes from the change in flow direction, from a small to a larger, more perpendicular angle to the topography. The model simulation for the flow at 12.5 m s$^{-1}$ from
195° (Fig. 11a) is not unlike the result of 7.5 m s$^{-1}$ and 210° (cf. Fig. 10b), but without the rainfall by sea breeze over the central plains. When the $F_{rw}$ increases from almost zero to 0.26 as the prevailing flow is turned at a larger angle (but at a



slower speed), the model produces less and less total rainfall in Taiwan (Figs. 11a-c), presumably due both to a suppression to local circulation and an increase of blocking effect. This tendency, although not as apparent, also exists in the observation (Figs. 11d-f). However, similar to the low-$F_{rw}$ conditions in Fig. 10, the model still produces too much rainfall in the eastern

sub-region but not enough in central Taiwan. In Fig. 11d, the observed rainfall in southwestern Taiwan might be caused by migratory rainfall systems from upstream (i.e., the northern South China Sea) as the flow was close to southerly.

**5.3 Flow from low-$F_{rw}$ to high-$F_{rw}$ regime**

The final three cases of model and observed results are shown in Fig. 12, and the $F_{rw}$ increases from 0.12 to 0.56, therefore from the low-$F_{rw}$ to high-$F_{rw}$ regime. The flow directions are at a considerable angle to the terrain, at 12.5 m s$^{-1}$ from 210°

and 240°, and 20 m s$^{-1}$ from 240°, respectively. Some of these conditions are similar to the disturbed periods in Ruppert et al. (2013) but before frontal arrival, with rainfall mainly over the mountains. In model results, the three cases are identified as low-$F_{rw}$, mixed, and high-$F_{rw}$ regime (cf. Table 4), and the rain-producing mechanisms are island circulation, mixed, and topographic uplift, following the order. This change in rainfall mechanism is evident in the model (Figs. 12a-c) as well as in the observation (Figs. 12d-f). Not only in pattern, the dramatic increase in rainfall amount in the high-$F_{rw}$ regime when the

southwesterly flow reaches 20 m s$^{-1}$, with maxima in mountain interiors of central and southern CMR, is well exemplified in Fig. 12f. While the event on 11-12 June 2012 was extreme and rare in northern Taiwan (e.g., Wang et al., 2016), the overall distribution in the mountains highly resemble the climatology (cf. Fig. 2a), and asserts the dominant role of the large events toward the total rainfall in the Mei-yu season. In Fig. 12f, considerably more rainfall was observed in the real event compared to the idealized model result (cf. Fig. 12c), since the near-surface moisture content was higher in reality. However,

the rainfall in northern Taiwan was linked to the Mei-yu front (Wang et al., 2016), a mechanism not existent in the model simulations. Existing disturbances might also be responsible for the rainfall in southwestern Taiwan in Fig. 12d and that in northwestern Taiwan in Fig. 12e. For the second case (12.5 m s$^{-1}$ and 240°), interestingly, the observed rainfall was also much more than in the model, although the moisture content was lower in the observation (Figs. 12b,e). Overall, it is found that many responses in rainfall distributions in the model as the flow conditions are changed can be applied to the real

atmosphere with a similar tendency, but some discrepancies nevertheless also exist.

**6 Conclusion and summary**

In this work, the rainfall regime and characteristics in Taiwan during the Mei-yu season are studied through idealized simulations using the CReSS model at a grid size of 2 km, under prescribed wind direction and speed combinations of southwesterly flow in the lower to middle troposphere but with real topography and diurnal effects. Thus, compared to

earlier idealized studies (e.g., Chu and Lin, 2000; Chen and Lin, 2005a,b; Sever and Lin, 2017), both the dynamic and thermodynamic roles played by the topography can be isolated and examined without the influence of Mei-yu front or other disturbances commonly found in real events. Based on averaged and smoothed sounding profile, three-dimensional (3D)



idealized flow fields are constructed using the geostrophic wind relationship in the free atmosphere, and modified to take into account friction inside the PBL. Eight wind directions from 180° to 285° every 15° and eight wind speeds from 5 to 22.5

m s$^{-1}$ every 2.5 m s$^{-1}$, giving a total of 64 combinations of prescribed flows that are fixed over 950-500 hPa. Then, these horizontally uniform and steady fields are provided to the CReSS model as IC/BCs for integration of 50 h, in which the first two hours are for spin-up and excluded from analysis. The rainfall amounts and patterns under different flow (and wet Froude number, or $F_{rw}$) conditions are analyzed to illustrate the role of Taiwan's topography. To investigate on the effects of moisture content inside the PBL, several RH values are specified (from 70% to 100% every 7.5% plus 55%) for nine of the

64 combinations (210°, 240° and 270° for wind direction, and 10, 15, and 20 m s$^{-1}$ for wind speed), thus another 45 experiments are carried out for this purpose.

From the model results in response to different southwesterly wind direction and speed combinations, where the RH near the surface is set to 85% (based on the mean sounding) and corresponds to a CAPE of ~2800 J kg$^{-1}$, three rainfall regimes in Taiwan with different range of $F_{rw}$ are identified. The first regime is the low-$F_{rw}$ regime, where the wind speed is typically

no more than 10 m s$^{-1}$, or at a higher speed but small angle to the topography, with a $F_{rw}$ about 0.3 or less. In this regime reminiscent to the undisturbed periods of Ruppert et al. (2013), the island circulation from thermodynamic effects (including upslope winds and see breeze) during daytime is the main cause of rainfall, which exhibits a pronounced diurnal cycle (in local afternoon). Under such conditions, the lower the prevailing wind speed, the more rainfall there is. When the flow speed is higher but more parallel to the terrain, the rainfall tends to reduce in amount and spatial coverage, and shift toward the

leeside area (from southern toward northern/eastern Taiwan).

The second regime is the (relatively) high-$F_{rw}$ regime when the flow is at least 12.5 m s$^{-1}$ and impinging on the topography at a large angle, as $F_{rw}$ is mostly ≥ 0.35-0.4 and can be up to 0.91 (flow at 22.5 m s$^{-1}$ from 285°). Under such conditions, the flow-over scenario takes place and topographic rainfall becomes dominant through mechanical uplift of unstable air, with rainfall maxima over the windward slopes of the mountains (or near the ridge) in southern and central Taiwan. While some

conditions in this regime are not unlike those in Chen and Chen (1995), Li and Chen (1998), and the disturbed periods of Ruppert et al. (2013), the most rainfall (peaking at 578 mm per day) occurs around the direction from 240°-255° at a given speed, and the rain (and $F_{rw}$) also increases with speed. It is also found that as the flow turns more perpendicular (from 255° to 285°) to the elongated mesoscale topography of Taiwan, the rainfall tends to decrease due to s stronger blocking effect, as found in previous studies (e.g., Wang et al. 2005), even though the value of $F_{rw}$ increases. As the dynamical effect of terrain

uplifting becomes more evident, the rainfall also becomes more persistent throughout the day with a reduced range of diurnal variations.

Between the two above rainfall regimes, there exists a third and mixed regime, with intermediate $F_{rw}$ values and rainfall characteristics in transition. Not identified in previous studies, such a mixed regime occurs at high speed (≥17.5 m s$^{-1}$) but small angles with $F_{rw}$ about 0.15-0.25, or at medium speed (10-15 m s$^{-1}$) but larger angles with $F_{rw}$ around 0.3-0.4. In the

three above regimes, comparison between selected cases with observations indicate that many responses in rainfall



distributions in the model as the flow conditions are changed can be applied to the real atmosphere with a similar tendency, but some discrepancies also exist.

For the effects of moisture content inside the PBL, the results indicate an increase (decrease) in overall and peak rainfall amount when the RH is increased (reduced) from the control value of 85%, as expected intuitively. However, the near-surface RH also plays a role to affect the instability and how easily the convection can be triggered in the model (easier in an environment with a higher RH inside the PBL). Thus, not only the amount of rainfall but also specific details such as where the peak rainfall would occur in Taiwan can be affected by the RH value in our tests. Overall, Taiwan's topography plays an important role in determining the rainfall amounts, distributions, and characteristics, even under idealized southwesterly flow conditions with prescribed direction and speed.

**Code and data availability**

The CReSS model and its user's guide are publicly available at http://www.rain.hyarc.nagoya-u.ac.jp/~tsuboki/cress_html/index_cress_eng.html. The sounding and other data needed to reproduce our results are being prepared, and will be stored in a data bank for public access.

**Author contribution**

C.-C. Wang developed the research idea, formulate its aims, designed the experiments, helped with the simulations, analysis and interpretation, provide funding and project administration, and prepared the manuscript with contributions from all co-authors. P.-Y. Chuang and S.-T. Chen performed the simulations and analysis. D.-I. Lee contributed to the research idea, design of experiments, and funding. K. Tsuboki created and provided the model code, and helped with the simulations.

**Competing interests**

The authors declare that they have no conflict of interest.

**Acknowledgements**

The authors would like to thank the anonymous reviewers for their valuable comments. All observational data and rainfall plots in Fig. 2a and those used in Figs. 10-12 are provided by the CWB. This study is supported by the Ministry of Science and Technology (MOST) of Taiwan, jointly under grants MOST 103-2119-M-003-001-MY2, MOST 105-2111-M-003-003-MY3, MOST 108-2111-M-003-005-MY2, MOST 110-2111-M-003-004, and MOST 110-2625-M-003-001.





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



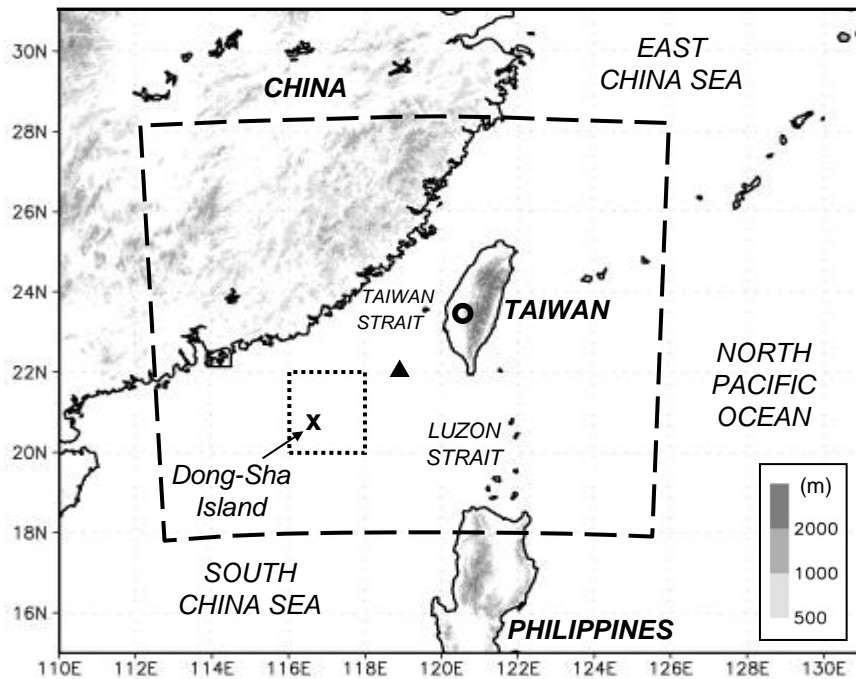

**Figure 1: The geography and topography (m, shading) surrounding Taiwan. The dashed lines show the CReSS model simulation domain, and the locations of Dong-sha Island (cross), research vessel (triangle), and the reference point of the idealized initial and boundary conditions (open circle, at 23.5°N, 120.5°E) are all marked. The dotted box depicts the 2° × 2° area surrounding Dong-sha used to compute mean wind.**



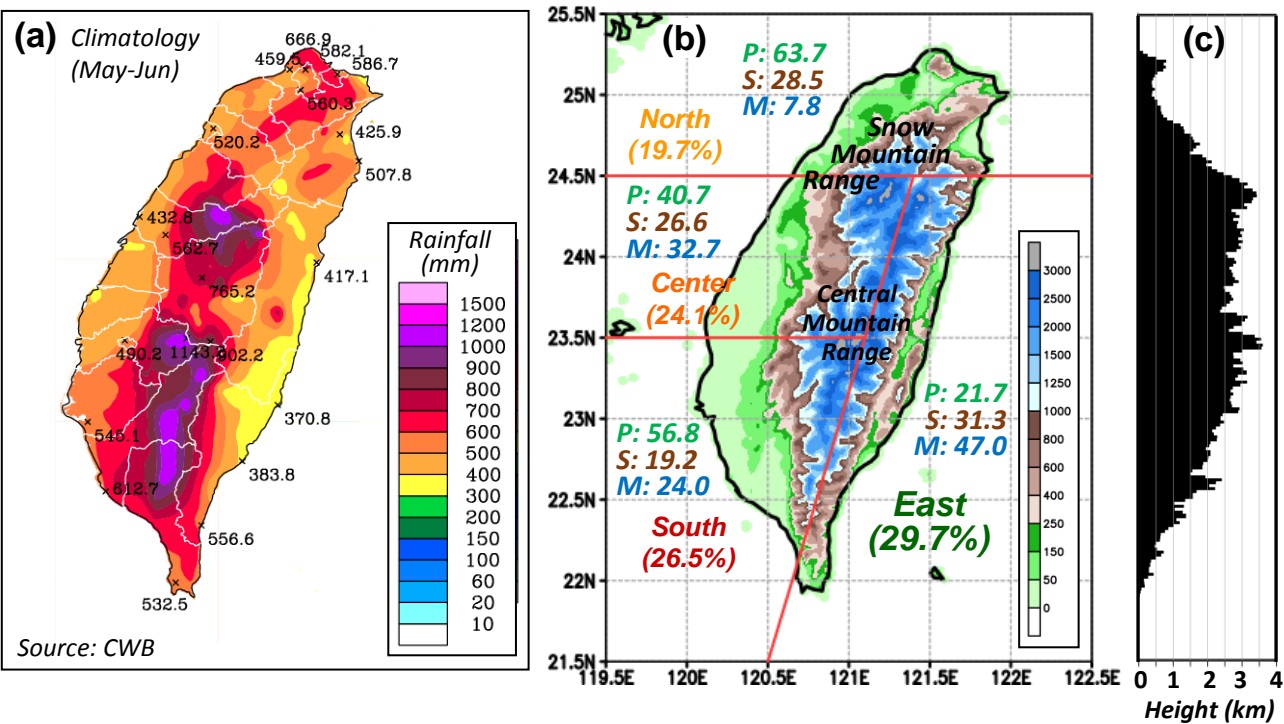

**Figure 2: (a) The distribution of total accumulated rainfall (mm) per mei-yu season (May-Jun) in the climatology of 1981-2010 (source: CWB). (b) The topography (m, color) of Taiwan. The three elevation ranges of plain (< 250 m), slope (0.25-1 km), and mountain (≥1 km) and the four sub-regions (north, center, south, and east) also shown with their percentages (%), and (c) the**
**north-south profile of the highest topography (km).**


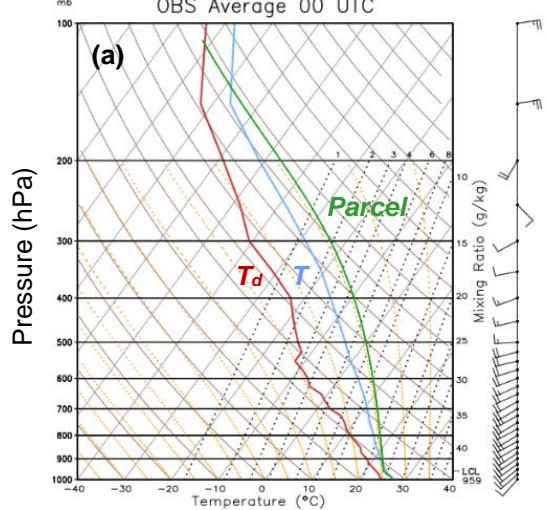

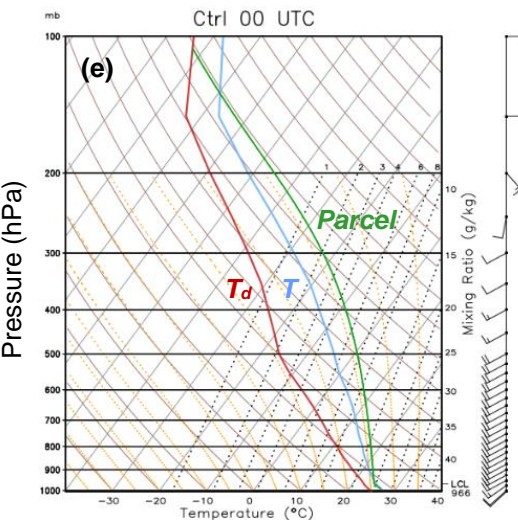

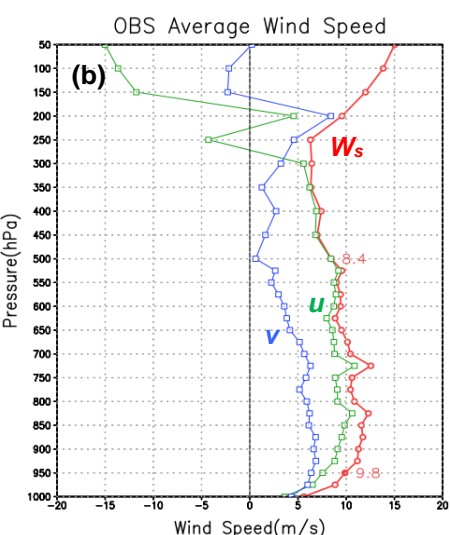

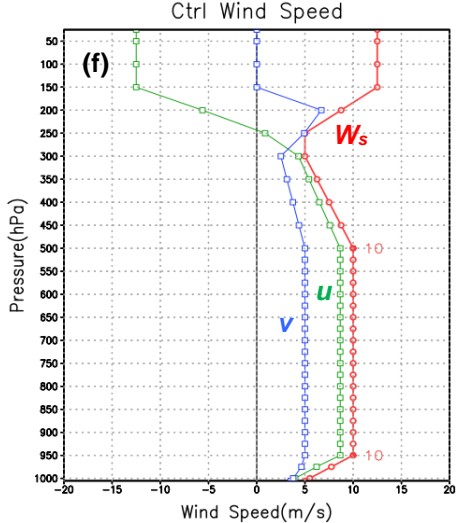


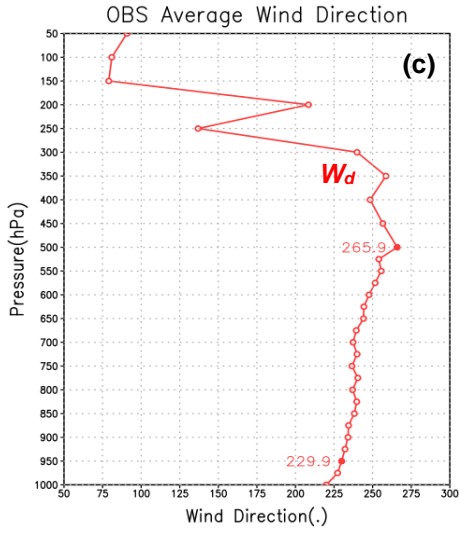

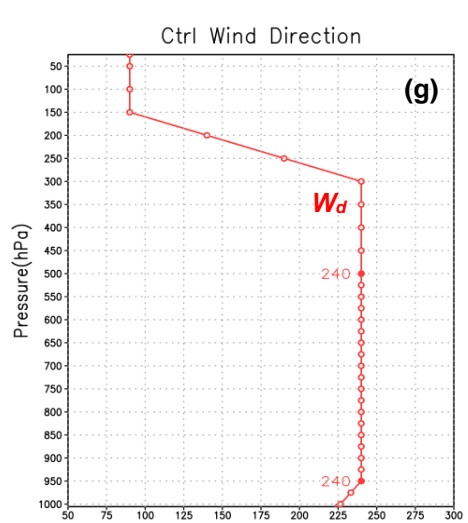

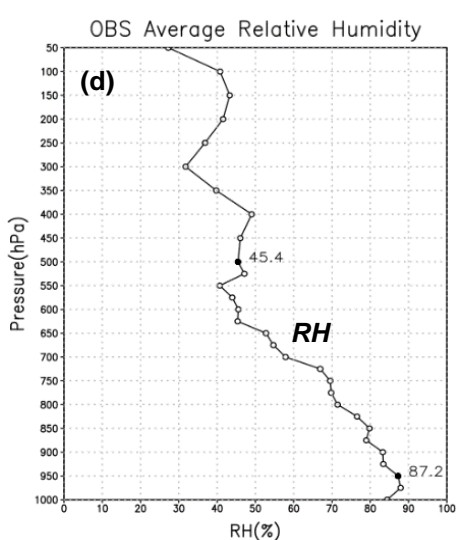

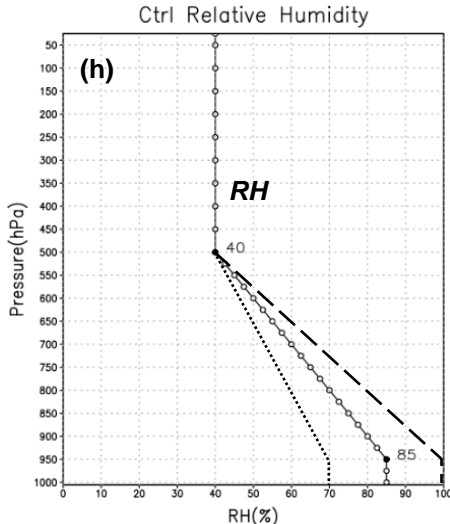

**Figure 3: Mean vertical profiles of (a) temperature ($T$, °C), dew-point temperature ($T_d$, °C), and wind (kt), together with the process curve for a surface parcel (following dry/moist adiabatic motion) in the Skew $T$-log $p$ diagram, horizontal wind (m s$^{-1}$), including (b) $u$ and $v$ components and wind speed ($W_s$, all in m s$^{-1}$) and (c) wind direction ($W_d$, °), and (d) relative humidity (RH, %) from seven soundings taken at Dong-sha Island and the Research Vessel (RV) upstream from Taiwan (see text for details). (e)-(h) As in (a)-(d), except for the smoothed or prescribed profiles used in the CTL experiment (see text for details). In (h), the long-dashed and dotted lines depict two other RH profiles used in moisture tests, with RH = 100% and 70% below 950 hPa, respectively (no difference from CTL at and above 500 hPa).**







**Figure 4: The distributions of sea-level pressure (hPa, isobars, every 1 hPa; ocean only), surface wind (m s$^{-1}$, wind barbs; half barb**
**= m s$^{-1}$ and full barb = 10 m s$^{-1}$), and hourly rainfall (mm, color) at intervals of 3-5 h from (a) 0 h to (l) 43 h in the CTL experiment. The height contours at 0.25 and 1 km are also drawn over land (gray contours).**

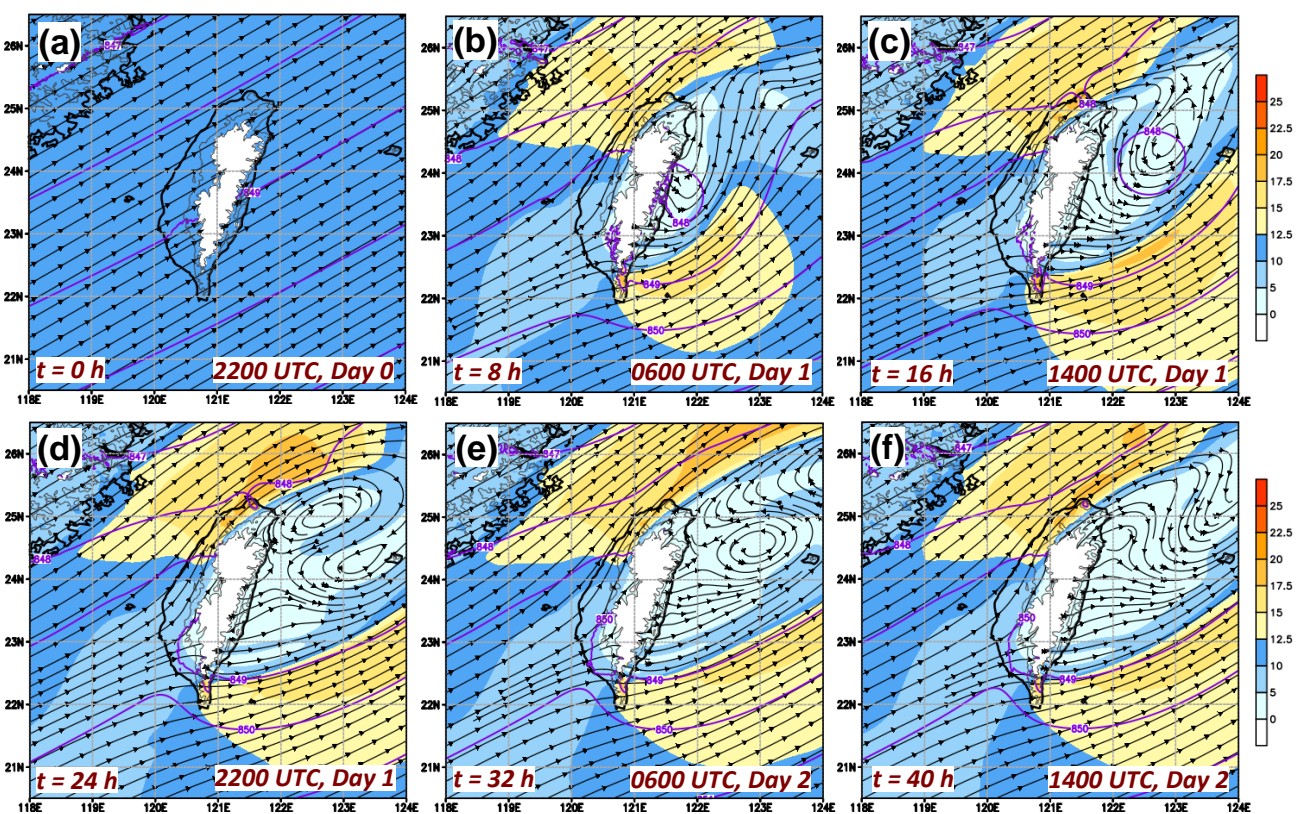

**Figure 5: The distributions of pressure (hPa, isobars, every 1 hPa), streamlines, and wind speed (m s⁻¹, color) at the height of 1481 m every 8 h from (a) 0 h to (f) 40 h in the CTL experiment. The height contours at 1481 m (gray) are also drawn.**



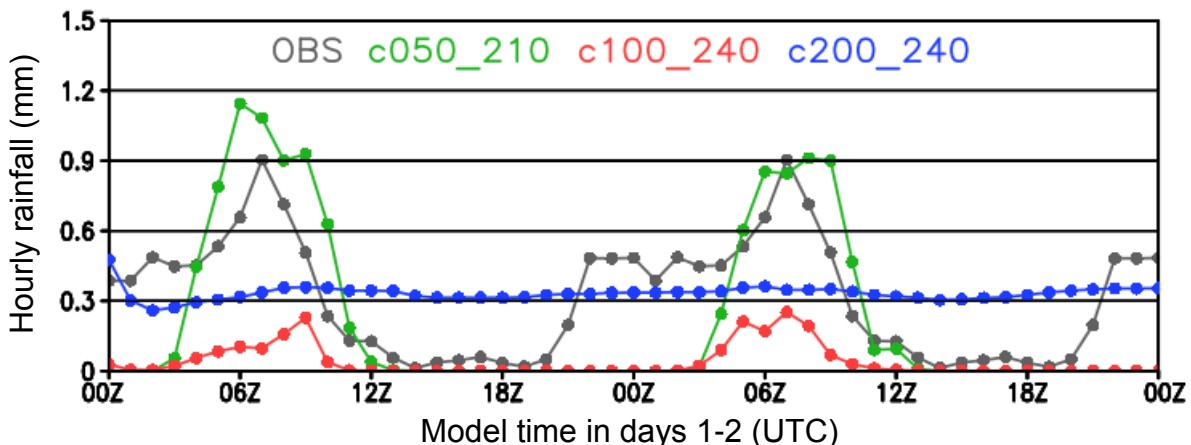

**Figure 6: The time series of spatially-averaged hourly rainfall (mm) over Taiwan in the observation (gray) and three model experiments: with uniform southwesterly winds at 5 m s⁻¹ from 210° (green), at 10 m s⁻¹ from 240° (red, i.e., the CTL), and at 20 m s⁻¹ from 240° (blue), respectively. The observation is the mean diurnal cycle (repeated twice) from the dates of the sounding data (see text for detail).**







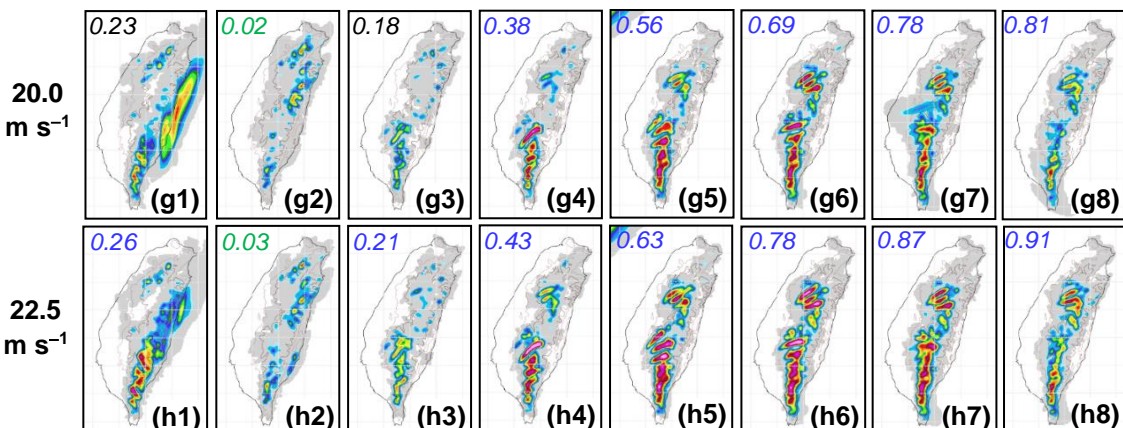

**Figure 7: Mean daily rainfall distribution (mm, per 24 h, scale on the right) over Taiwan in the 8 × 8 experiments of different wind direction (every 15° from 180° to 285°) and speed (every 2.5 m s⁻¹ from 5.0 to 22.5 m s⁻¹), as labeled on top (for direction) and left (for speed) from (a1) to (h8), respectively. The CTL experiment (240°, 10 m s⁻¹) is shown in (c5) using a red box. The value of $F_{rw}$ is labelled on the upper-left corner in each panel, and blue, green, and black colors indicate the rainfall regime of terrain uplift, island circulation, and mixed, respectively.**




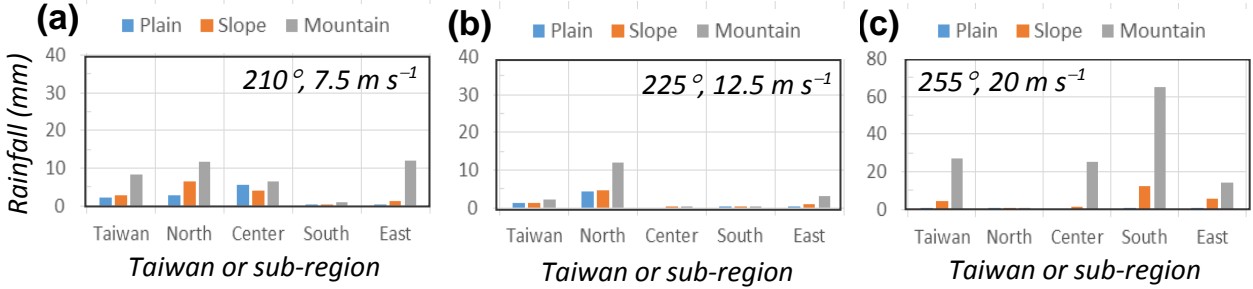

**Figure 8: Spatial-averaged mean daily rainfall (mm) at the three elevation ranges (plain, slope, and mountain) over Taiwan and its four sub-regions in three experiments, with uniform southwesterly flow (a) from 210° at 7.5 m s⁻¹, (b) from 225° at 12.5 m s⁻¹, and (c) from 255° at 20 m s⁻¹, respectively. Note the difference in the scale of vertical axes.**

Natural Hazards
and Earth System


**Figure 9: As in Fig. 7, but for the four sets of 3 × 3 experiments of different near-surface RH of (a) 100%, (b) 92.5%, (c) 77.5%,**
**and (d) 70%, respectively. The wind direction (210°, 240°, or 270°) and speed (10, 15, or 20 m s⁻¹), and the value of $F_{rw}$ are all**
**labelled.**

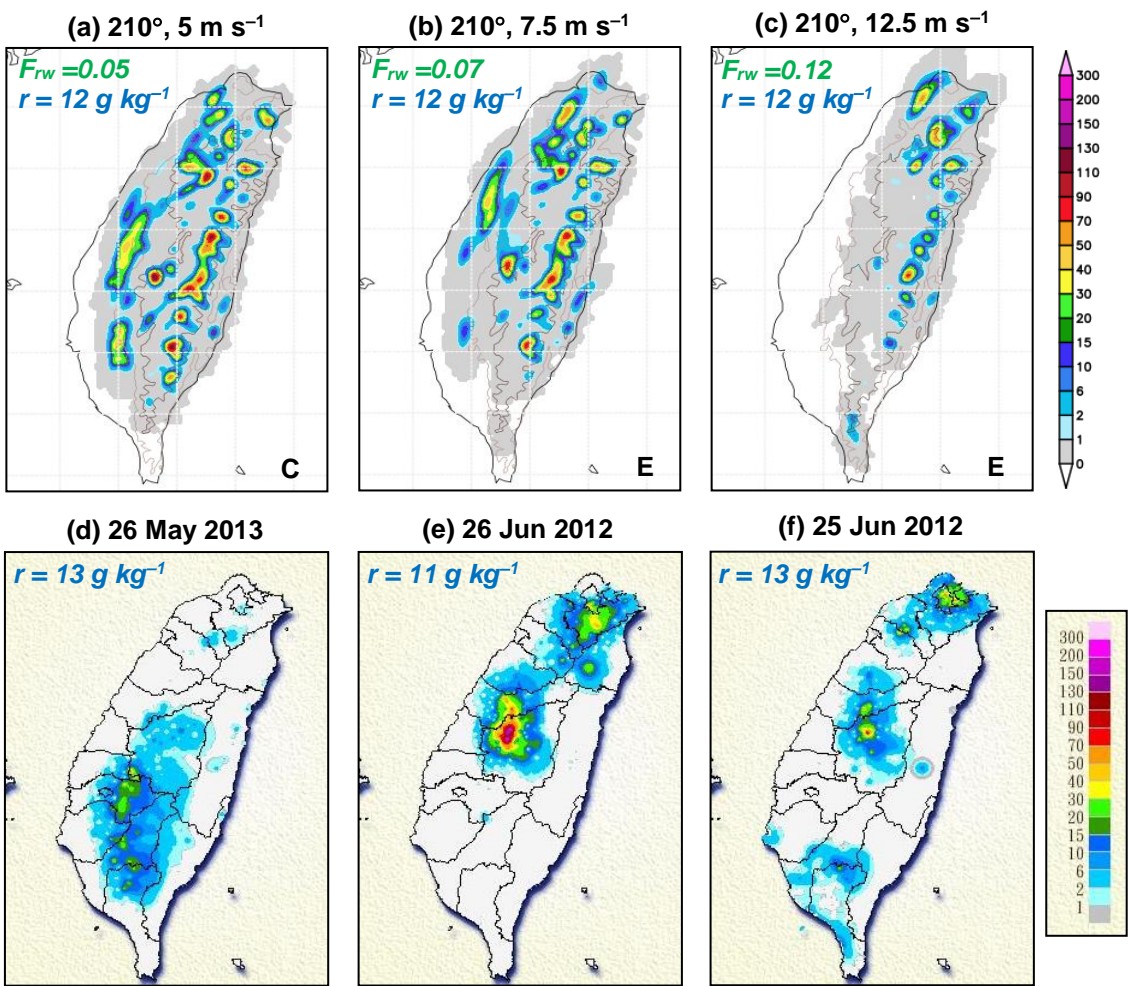

**Figure 10: Comparison between results of idealized daily rainfall distributions (mm) in this study (top row) and real events of specific dates in observation (bottom row, source: CWB), for the flow direction of 210° in the low-$F_{rw}$ regime. The three idealized wind speeds include (a) 5 m s⁻¹, (b) 7.5 m s⁻¹, and (c) 12.5 m s⁻¹, and the corresponding dates in observation are (d) 26 May 2013, (e) 26 Jun 2012, and (f) 25 Jun 2012, respectively. The values of $F_{rw}$ and near-surface mixing ratio (g kg⁻¹, top left) as well as the sub-region of peak daily rainfall (N, C, S, or E; lower right) in the idealized results are all labelled.**
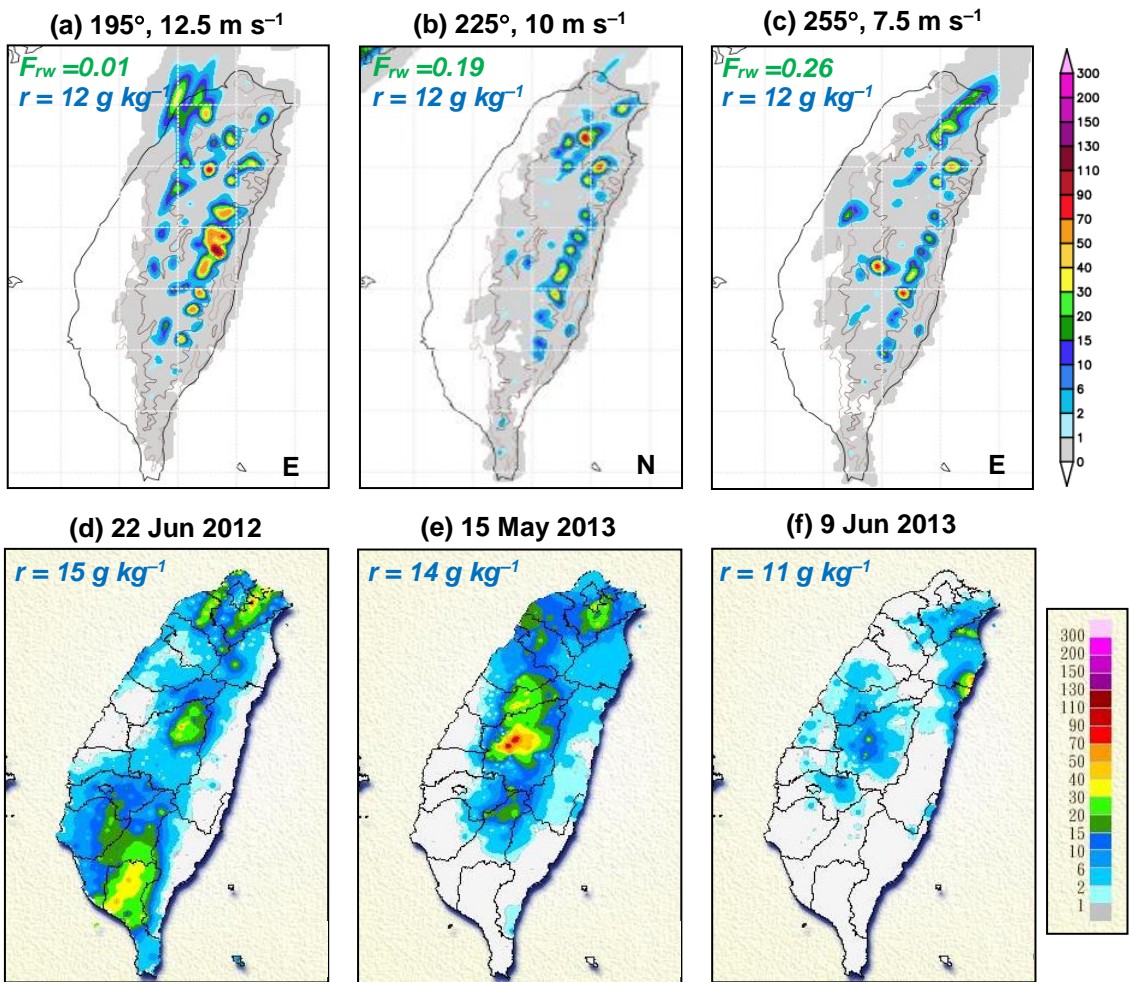

**Figure 11: As in Fig. 10, but between idealized results and observations (source: CWB), for different flow directions from more parallel to more perpendicular in the low-$F_{rw}$ regime. The three idealized cases are (a) 12.5 m s$^{-1}$ from 195°, (b) 10 m s$^{-1}$ from 225°, and (c) 7.5 m s$^{-1}$ from 255°, and the corresponding dates in observation are (d) 22 Jun 2012, (e) 15 May 2013, and (f) 9 Jun 2013, respectively.**

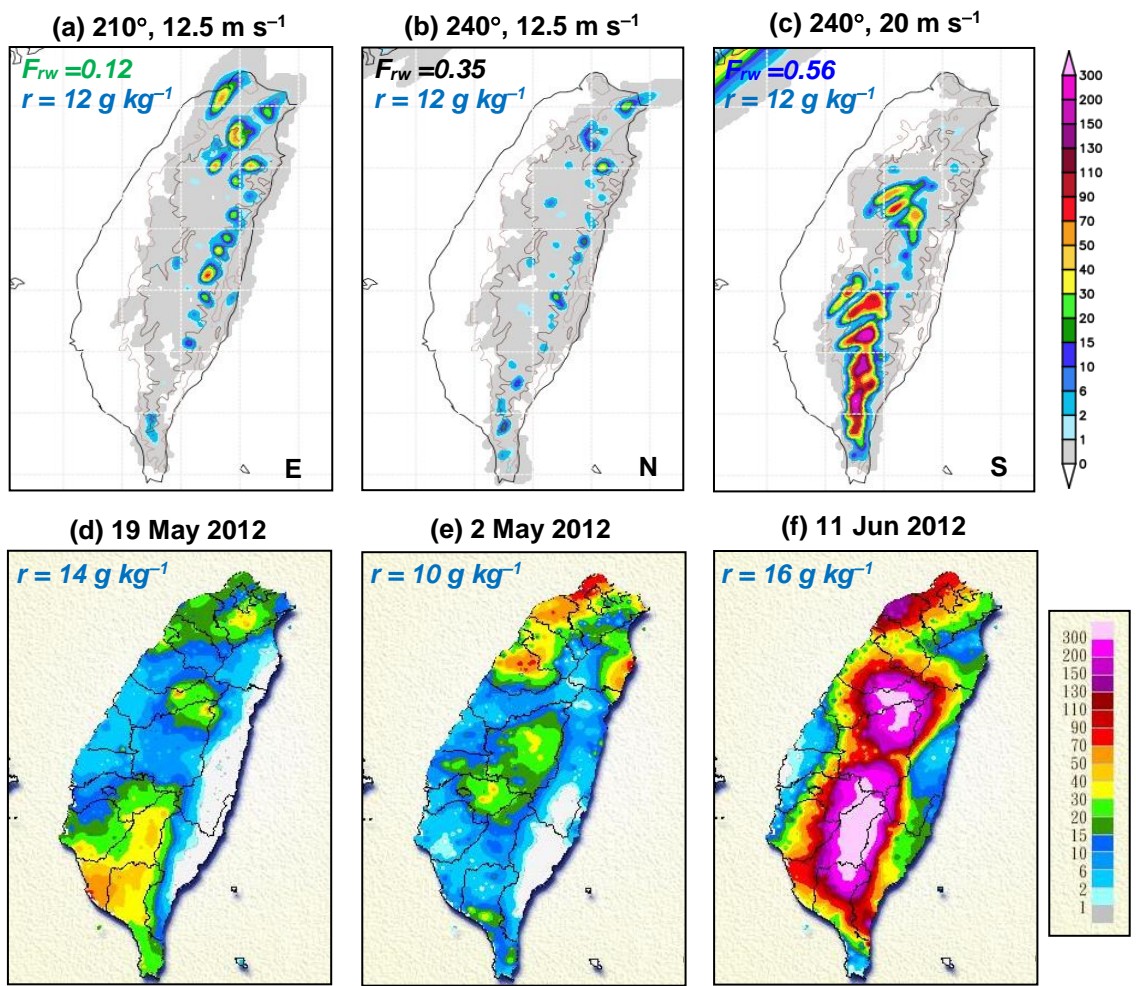

**Figure 12: As in Fig. 10, but between idealized results and observations (source: CWB), for flow directions from 210° to 240° at an increased speed, thus from a low-$F_{rw}$ to high-$F_{rw}$ regime. The three idealized cases are (a) 12.5 m s⁻¹ from 210°, (b) 12.5 m s⁻¹ from 240°, and (c) 20 m s⁻¹ from 240°, and the corresponding dates in observation are (d) 19 May 2012, (e) 2 May 2012, and (f) 11 Jun 2012, respectively.**




| Wind profile in the vertical for a prescribed southwesterly flow: | |
|---|---|
| 950-500 hPa | Fixed at the prescribed direction/speed (e.g., 240°/10 m s$^{-1}$) |
| At 300 hPa and above | Fixed at the modified sounding (Figs. 3f,g, same for all runs) |
| 500-300 hPa | Linearly interpolated between winds at 500 and 300 hPa |
| Surface to 950 hPa | Linearly reduced (from 950 hPa) to half the speed and 15° to the left at the surface |
| Moisture profile in the vertical for a prescribed near-surface RH value: | |
| Surface to 950 hPa | Fixed at the prescribed RH value (e.g., 85%) |
| At 500 hPa and above | Fixed at 40% (as in Fig. 3h, same for all runs) |
| 950-500 hPa | Linearly interpolated between RH values at 950 and 500 hPa |

**Table 1: The methods used to construct the idealized wind and moisture profiles (at the reference point of 23.5°N, 120.5°E) in this study.**

| | |
|---|---|
| Projection | Lambert Conformal (center at 120°E, secant at 10°N and 40°N) |
| Grid spacing ($x$, $y$, $z$; km) | 2 × 2 × 0.1-0.62 (0.4)* |
| Grid dimension ($x$, $y$, $z$) and domain size (km) | 660 × 560 × 50 (1320 × 1120 × 20) |
| IC/BCs | Idealized 3-D data (0.25° × 0.25°, 32 levels) |
| Topography | Digital elevation model at (1/120)° |
| Sea surface temperature | NOAA mean SST analysis (1° × 1°) for May-Jun 2008 |
| Initial time, integration length, and output frequency | 2200 UTC (0600 LST), 50 h, 1 h |
| Cloud microphysics | Bulk cold-rain scheme (6 species) |
| PBL parameterization | 1.5-order closure with prediction of turbulent kinetic energy |
| Surface processes | Energy/momentum fluxes, shortwave and longwave radiation |
| Substrate model | 41 levels, every 5 cm to 2-m deep |

740 **Table 2: The setup of CReSS model domain, IC/BCs, and physical schemes in this study. * The vertical grid spacing (Δz) of CReSS is stretched (smallest at the bottom), and the averaged spacing is given in the parentheses.**





| Direction \ Speed | 180° | 195° | 210° | 225° | 240° | 255° | 270° | 285° |
|---|---|---|---|---|---|---|---|---|
| 5.0 | 0.06 | 0.01 | 0.05 | 0.10 | 0.14 | 0.17 | 0.19 | 0.20 |
| 7.5 | 0.09 | 0.01 | 0.07 | 0.14 | 0.21 | 0.26 | 0.29 | 0.31 |
| 10.0 | 0.11 | 0.01 | 0.09 | 0.19 | 0.28 | 0.35 | 0.39 | 0.41 |
| 12.5 | 0.14 | 0.01 | 0.12 | 0.24 | 0.35 | 0.44 | 0.49 | 0.51 |
| 15.0 | 0.17 | 0.02 | 0.14 | 0.29 | 0.42 | 0.52 | 0.58 | 0.61 |
| 17.5 | 0.20 | 0.02 | 0.16 | 0.33 | 0.49 | 0.60 | 0.68 | 0.71 |
| 20.0 | 0.23 | 0.02 | 0.18 | 0.38 | 0.56 | 0.69 | 0.78 | 0.81 |
| 22.5 | 0.26 | 0.03 | 0.21 | 0.43 | 0.63 | 0.78 | 0.87 | 0.91 |

Table 3: The values of moist Froude number ($F_{rw}$) in the experiment set with prescribed southwesterly wind direction (°, column) and speed (m s$^{-1}$, row). Cells filled with light gray indicate a rainfall regime mainly from island circulation (with lower $F_{rw}$), and those filled with medium gray indicate a regime dominated by terrain uplift (with higher $F_{rw}$), based on simulated daily rainfall pattern over Taiwan. The cells with no color indicate mixed rainfall from both mechanisms.

| Direction \ Speed | 180° | 195° | 210° | 225° | 240° | 255° | 270° | 285° |
|---|---|---|---|---|---|---|---|---|
| 5.0 | **5.31 (159)** | **6.31 (183)** | **5.49 (151)** | **5.02** (144) | 4.43 (140) | 4.21 (**173**) | 3.85 (**164**) | 4.04 (**155**) |
| 7.5 | 4.50 (**170**) | **5.64** (116) | 4.19 (111) | 3.38 (116) | 2.12 (138) | 1.61 (96) | 1.32 (108) | 1.44 (117) |
| 10.0 | 4.34 (**163**) | 4.62 (**188**) | 2.60 (109) | 1.64 (112) | 0.92 (58) | 0.44 (43) | 0.45 (43) | 0.55 (72) |
| 12.5 | 2.96 (107) | 3.24 (149) | 1.66 (87) | 1.56 (95) | 0.55 (44) | 0.32 (26) | 0.34 (23) | 0.36 (26) |
| 15.0 | 1.85 (114) | 2.78 (**165**) | 0.87 (69) | 2.06 (**156**) | 0.83 (55) | 0.90 (51) | 0.90 (36) | 0.35 (20) |
| 17.5 | 1.76 (104) | 2.07 (147) | 0.75 (60) | 2.12 (132) | 2.98 (**189**) | 3.80 (**155**) | 3.02 (121) | 0.88 (48) |
| 20.0 | 3.33 (103) | 1.65 (84) | 1.27 (52) | 4.78 (**340**) | **8.06 (342)** | **9.37 (283)** | **6.51 (254)** | 2.73 (139) |
| 22.5 | 5.67 (**316**) | 1.59 (89) | 2.19 (82) | **8.86 (549)** | 13.06 (**578**) | 14.76 (**512**) | 10.28 (**330**) | **5.29 (223)** |

Table 4: The daily mean rainfall over Taiwan (mm) and the peak amount (parentheses) in the experiment set with prescribed southwesterly wind direction (°, column) and speed (m s$^{-1}$, row). The areal-mean (peak) values ≥ 5 (150) mm are in boldface. Cells filled with light, medium, and dark gray and no color indicate that the peak amount (all in mountain elevation) occurs in the sub-region of northern, central, southern, and eastern Taiwan (cf. Fig. 2b), respectively.




| Relative humidity (RH, %) from surface to 950 hPa | CAPE (J kg⁻¹) |
|---|---|
| 100.0 | 5546 |
| 92.5 | 4148 |
| 85.0 | 2803 |
| 77.5 | 1521 |
| 70.0 | 464 |
| 55.0 | 0 |

**Table 5: The CAPE values of experiments with different near-surface moisture content.**

| Wind direction | RH / Speed | 55.0 | 70.0 | 77.5 | 85.0 | 92.5 | 100.0 |
|---|---|---|---|---|---|---|---|
| 210° | 10.0 | 39 (M) | 96 (M) | 132 (M) | 109 (M) | 117 (M) | 114 (M) |
|  | 15.0 | 20 (M) | 46 (M) | 51 (M) | 69 (M) | 85 (M) | **245** (S) |
|  | 20.0 | 3 (M) | 12 (M) | 21 (M) | 52 (M) | **168** (M) | **465** (S) |
| 240° | 10.0 | 13 (M) | 29 (M) | 60 (M) | 58 (M) | 64 (M) | **225** (P) |
|  | 15.0 | 6 (M) | 17 (M) | 24 (M) | 55 (M) | 134 (M) | **433** (M) |
|  | 20.0 | 22 (M) | 134 (M) | **209** (M) | **342** (M) | **749** (M) | **994** (M) |
| 270° | 10.0 | 11 (M) | 23 (M) | 29 (M) | 43 (M) | 90 (M) | **123** (S) |
|  | 15.0 | 1 (M) | 11 (M) | 21 (M) | 36 (M) | 107 (M) | **213** (M) |
|  | 20.0 | 19 (M) | 84 (M) | **153** (M) | **254** (M) | **337** (M) | **384** (M) |

755 **Table 6: As in Table 4, but for the daily peak rainfall over Taiwan (mm) in all the experiment set to test the effects of near-surface moisture. The results for different wind speed (m s⁻¹) and RH (%) from top to bottom are those with a wind direction from 210°, 240°, and 270°, respectively. The background colors of cells and boldface have the same meaning in sub-region as in Table 4, and the parenthesis gives the elevation range (P, S, or M).**