# Peer review of "Idealized Simulations of Mei-yu Rainfall in Taiwan under Uniform Southwesterly Flow using A Cloud-Resolving Model"

_Natural Hazards and Earth System Sciences, 2021_

## Referee Comment (RC1)

Review of

Idealized Simulations of Mei-Yu Rainfall in Taiwan under uniform Southwesterly Flow Using a Cloud-Resolving Model

By Wang et al.

The authors used a cloud-resolving model to investigate the role of the complex topography in Taiwan on rainfall characteristics during the Mei-Yu season without the influence of fronts or disturbances.  They initialize the model using horizontally uniform flow without vertical shear with different wind speeds and directions.  They characterized their rainfall regimes based on the wet Froude number ($Frw$). For the low-$Frw$ regime, rainfall production is dominated by thermal forcing from the surface, whereas for the high $Frw$ regime, the mechanical uplift of unstable air becomes important.  Between these two regimes, the mixed regime exists for intermediate $Frw$ number.  They also compare their model results with real cases in Section 5.  The manuscript is fairly well written.  However, there are significant major concerns the authors will need to address before this work can be accepted for publication.

1. This study is lacking well-defined scientific objectives.  It is well-known that under weak wind conditions, thermally driven diurnal circulations are important, whereas under strong wind conditions with a large impinging angle, mechanical uplift becomes significant.  What is new?

2.  The Froude number has been used to classify flow regimes (flow over vs blocked flow regimes) for airflow over an isolated mountain by many authors.  Compared with classical theoretical studies, that use a bell-shaped mountain, this study uses the real-terrain of Taiwan.   However, the results from a series of numerical experiments in this study simply confirm this well known fact.

3. The authors fail to state the theoretical basis or hypothesis to invoke Froude number theory for the rainfall regimes.

4. In Section 5, the authors attempt to compare their model simulations initialized by a single upstream sounding to real cases of heavy rainfall events during the Mei-Yu season over Taiwan.   This is simple minded.  Heavy rainfall events in many different parts of the world are related to synoptic and mesoscale processes in addition to orographic effects (G. Chen 1983, JMSJ; Doswell, 1987, WAF; Doswell et al., 1996, WAF; Maddox et al., 1979, BAMS; and many others).  Without including these processes in the models, it is unlikely that numerical simulations initialized by a single sounding will be able to simulate these events.

 5. The authors state that the wet $Fr$ number is very close to the $Fr$ number.  Thus, in terms of flow regimes, the moisture is not important.  However, for heavy rainfall events, moisture availability is a significant parameter for rainfall production and may be more important than the variations in the Froude number for $Fr < 1$.

6. The upstream sounding used is horizontally uniform with very little vertical wind shear below the 500-hPa level. Is this a typical sounding for heavy rainfall events over Taiwan?  Do soundings in the warm sector of Mei-Yu systems exhibit clockwise turning with respect to height due to warm advection?

7. Except in the lowest levels, the thermodynamic profiles used seem rather dry.  Is this typical for the heavy rainfall soundings during the Mei-Yu season over Taiwan?

8. The authors use observed SST as the lower boundary condition over the open ocean.  The reviewer presumes there are spatial variations in SST in this region.  How would the spatial variations in SST affect the horizontal distributions of thermodynamic fields in the mixed layer and the depth of the mixed layer?  Are those being considered in the model initial conditions?

9.   Each simulation was run for 50 hours and the first two hours are considered as the model spin up period.  How is the initial spin up period determined?

10.  To address the effects of thermal forcing from the land surface, it is imperative to describe the lower boundary conditions over land used in the model.   The authors should also compare the simulated diurnal variations in temperature, winds and rainfall with observations very carefully.  Fig. 4 shows simulated rainfall on the eastern leeside in the afternoon hours which is odd.   It fails to show the effects of orographic uplift on rainfall production.

11. Areal averaged rainfall shown in Fig. 6 is inadequate.   There must be large spatial variations in rainfall throughout the diurnal cycle.   What are the days used for observations (gray curve) in Fig. 6?  How often do you observe southwesterly winds > 20 m s$^{-1}$?

Minor points:

1. The figure caption for Fig. 2b is very confusion.

2. Figure 4: Should provide information on local time.  The authors should also adjust the color table.

---

## Referee Comment (RC2)

**Review comments**
**nhess-2021-196**

**Title:** Idealized Simulations of Mei-yu Rainfall in Taiwan under Uniform Southwesterly Flow using A Could-Resolving Model

**Authors:** Wang et al.

**Recommendation:** major revisions

**General comments**

This study presents findings from idealized simulations for the island of Taiwan in which a uniform southwesterly flow is prescribed at fixed directions/speed combinations to investigate rainfall characteristics in the absence of large-scale frontal systems. In addition, near-surface relative humidity is varied and a subset of the simulations has been compared to observational data. The authors identify three rainfall regimes that correspond to different ranges of the wet Froude number and possible mechanisms for the resulting precipitation location and intensity are hypothesized. Although the paper is mostly well written and the illustrations have a good quality, it is a bit hard to see the innovation of this paper. The main result regarding the dominant process for rainfall production through mechanical uplift or thermal forcing is pretty much expected and there is a lack of evidence for their hypotheses. Although I like the general concept of idealized simulations using a real topography, I find the implementation and the connection to previous work for other islands unsuccessful so far. I would welcome a revised paper that is more physics-based, but that would probably involve substantial additional work and rewriting of the paper.

**Specific comments**

1. experimental design:

   - Why do you restrict the flow direction to SW? Is the southwesterly flow during that time of the year dominant? How often are there situations without the Mei-yu front? This important information needs to be given to determine if the model setup is representative or not.

   - Are the fine steps of 2.5 m/s and 15 degrees really necessary? Would not a larger range of flow direction with larger steps (e.g. 5 m/s and 30 degrees) be more informative? For example, in the study of Metzger et al. (2014), the incoming wind direction for the island of Corsica has been changed in steps of 30 degrees to cover the all possible wind directions. Although Corsica is a smaller island, these previous results should be cited in this study. Furthermore, within the framework of the HyMeX project, several other publications covering island convection and terrain effects were published.

     Metzger, J., C. Barthlott, N. Kalthoff (2014): Impact of upstream flow conditions on the initiation of moist convection over the island of Corsica, Atmos. Res. 145-146, 279-296, DOI:10.1016/j.atmosres.2014.04.011

- Why do you use an integration time of 50 h? Would not 24 h be sufficient? Which day do you take for the analyses in Fig. 7 etc.? Day 1, day 2 or the mean of both?

- What boundary conditions are applied in the model? Open, periodic?

- Deep convection is considered to be resolved at 2-km grid spacing, but is shallow convection still parameterized? If yes, how? Please specify.

2. I do not understand why the Froude number changes with the wind direction. If the wind speed does not change and the mountain height is constant as well, the Froude number should be independent of the flow direction unless the stability is changed. The authors should make an effort to explain how they calculate their Froude number in detail (spatial average, at what time, ...).

3. The authors speculate about the involved processes, i.e. terrain uplift and/or sea breeze/thermal circulations. None of these are assessed or proven in a quantitative way. Only for the CTL-run presented in Fig. 5, there is some evidence by the streamlines. I suggest to include additional material, e.g. low-level moisture convergence for establishing the impact of sea breeze on island convection.

4. Fig. 6: Observed precipitation starts to increase at around 20 UTC and reaches a plateau between 22–05 UTC before it further rises to the maximum value at 07 UTC. What mechanisms are reponsible for the plateau?

5. L71: What are "unwanted features"? Please specify.

6. The intercomparison to observations mostly shows a bad agreement between simulations and observational data (Fig. 10, 11, 12). Either the environmental conditions in the dates chosen do not match the model settings or other processes are missing in the model. I suggest to run realistic simulations with initial and boundary conditions from an operational model or other global analyses for these cases.

**Technical corrections**

- L15: local afternoon

- L17: This sentence needs to be rephrased. What is a "large angle"?

- L40: Blumen, 1990: Blumen is the book editor for Banta (1990). Do the authors mean the Banta article here?

- L45: orographic precipitation can often be resulted → please rephrase

- L60: Wang et al., 2002, 2003: For these years, there are only entries in the references for Wang and Chen (2002, 2003).

- L80: $Fr \to F_r$

- L144: Murakami et al. (1990, 1994) → Murakami (1990), Murakami et al. (1994)

- L147: Sagami → Segami

- L174: Chen and Lin (2005): Which entry is meant here, 2005a or 2005b?

- L184: The results of the CTL**-run** is ...

- L185: it  **behaves** as designed.

- L235: regime

- L330: ...this is resulted because... → please rephrase

- L565: Miguietta → Miglietta

---

## Author Comment (AC1)

NHESS-2021-196
Authors' Responses to Reviewer 1 (RC1, anonymous)
Date: 22 Nov 2021

Title: Idealized Simulations of Mei-yu Rainfall in Taiwan under Uniform Southwesterly Flow
    using A Cloud-Resolving Model
Authors: C.-C. Wang, P.-Y. Chuang, S.-T. Chen, D.-I. Lee, and K. Tsuboki

**1.  General comments:**

The authors used a cloud-resolving model to investigate the role of the complex topography in Taiwan on rainfall characteristics during the Mei-Yu season without the influence of fronts or disturbances. They initialize the model using horizontally uniform flow without vertical shear with different wind speeds and directions. They characterized their rainfall regimes based on the wet Froude number ($Frw$). For the low-$Frw$ regime, rainfall production is dominated by thermal forcing from the surface, whereas for the high $Frw$ regime, the mechanical uplift of unstable air becomes important. Between these two regimes, the mixed regime exists for intermediate $Frw$ number. They also compare their model results with real cases in Section 5. The manuscript is fairly well written. However, there are significant major concerns the authors will need to address before this work can be accepted for publication.

**Reply:** The constructive comments from this reviewer (Reviewer 1) are deeply appreciated, and the paper has been revised accordingly. In the revision (color-coded version), the changes made in response to Reviewer 1, Reviewer 2, and by ourselves (mostly some corrections and minor changes in English) are marked in red, blue, and orange, respectively. A point-by-point response to each of the comments from this reviewer are given below following their order. In each point, how and where the revision is made in the text is also specified.

**2.  Major comments:**

1)   This study is lacking well-defined scientific objectives. It is well-known that under weak wind conditions, thermally driven diurnal circulations are important, whereas under strong wind conditions with a large impinging angle, mechanical uplift becomes significant. What is new?

**Reply:** Thank you for the suggestion. In the revision, a well-defined scientific objectives is clearly stated in Section 1 (L97-99), as suggested. Compared to previous studies for Taiwan

in the literature, the main differences in the present study are the use of 3D framework with different wind directions, real topography, and also the inclusion of (diurnal) thermodynamic as well as the Coriolis effects. Thus, different rainfall regimes and the range of Froude number ($F_r$) for each of them can be identified with a better agreement with real conditions. This has not been done before for Taiwan. In the revision, the above arguments are better and more clearly conveyed to the readers (L88, L102-104, L215, L298-299), also as suggested.

2) The Froude number has been used to classify flow regimes (flow over vs blocked flow regimes) for airflow over an isolated mountain by many authors. Compared with classical theoretical studies, that use a bell-shaped mountain, this study uses the real-terrain of Taiwan. However, the results from a series of numerical experiments in this study simply confirm this well known fact.

**Reply:** Thank you for the suggestion. Compared to previous studies for Taiwan in the literature, the main differences in the present study are the use of 3D framework with different wind directions, real topography, and also the inclusion of thermodynamic (diurnal effects) as well as the Coriolis effects. Although the flow is still idealized, the results are more applicable and comparable to the observations at least to a reasonable extent (Section 5, please see our reply to major comment #4 below for detail description). In the revision, the above points are better clarified and conveyed to the readers (L88, L97-99, L102-104, L215, L298-299, L442), along the lines as suggested.

3) The authors fail to state the theoretical basis or hypothesis to invoke Froude number theory for the rainfall regimes.

**Reply:** As reviewed in Section 1, the Froude number theory has been applied and linked to different rainfall regimes in Taiwan in some earlier studies (e.g., Chen and Lin, 2005a,b), so the present study is not the first one to do so. However, in the revision, the general linkages between $F_r$ and rainfall regimes in Taiwan (orographic precipitation in high-$F_r$ regime and rainfall from island circulation and thermodynamic effects in low-$F_r$ regime) is better clarified (L53-54, L58-60), as suggested.

4) In Section 5, the authors attempt to compare their model simulations initialized by a single upstream sounding to real cases of heavy rainfall events during the Mei-Yu season over Taiwan. This is simple minded. Heavy rainfall events in many different parts of the world are related to synoptic and mesoscale processes in addition to orographic effects (G. Chen 1983, JMSJ; Doswell, 1987, WAF; Doswell et al., 1996, WAF; Maddox et al., 1979, BAMS; and many others). Without including these processes in the models, it

is unlikely that numerical simulations initialized by a single sounding will be able to simulate these events.

**Reply:** Thank you for this suggestion. The comparison between idealized simulations with real events of choice (when the conditions are relatively pure) in the previous draft was not successful, mainly because we didn't include the right data for comparison. In the revision, satellite cloud imageries at selected times are also provided (together with radar composite and the derived rainfall estimate), and they are much better to validate the model simulations (L194-195, L371, L374-378, L380-396, L402-414, L416-435, L565, L569-571, Figs. 11-13, p.36-38, L778-786, L789-791, L795-797), and the reasons why both the rain-gauge data (used in previous draft) and radar composites cannot capture the convection/rainfall along the eastern slopes of the CMR are provided (L383-387, L527-529, Fig. 2b, p.24, L707-711). At various places where needed, caveats are also added or stressed in the revision to clarify possible (or likely) differences between the model results and observations (L220-226, L365-369, L395-396, L408-414, L418-419, L424-426, L428-435), along the lines as suggested. For heavy-rainfall cases, previous modeling studies using gridded analyses and full physics are also cited in the revision (L425-426, L431-433).

5) The authors state that the wet *Fr* number is very close to the *Fr* number. Thus, in terms of flow regimes, the moisture is not important. However, for heavy rainfall events, moisture availability is a significant parameter for rainfall production and may be more important than the variations in the Froude number for *Fr* < 1.

**Reply:** Thank you for this comment and we agree. In the revision, it is clarified that $F_{rw}$ (and $F_r$) applies, strictly speaking, only to stable conditions with $N_w > 0$ (L190-191), and we also added that the moisture content near the surface affects the instability and rainfall production (L171), both as suggested.

6) The upstream sounding used is horizontally uniform with very little vertical wind shear below the 500-hPa level. Is this a typical sounding for heavy rainfall events over Taiwan? Do soundings in the warm sector of Mei-Yu systems exhibit clockwise turning with respect to height due to warm advection?

**Reply:** As described in Section 2.1, the sounding profile used in CTL is the mean from seven profiles at 0000 UTC and thus is typical in the Mei-yu season, but not necessarily conducive to heavy rainfall (as heavy rainfall occurred only in some of the sampled days). The rainfall information on these seven occasions are added as suggested (L117-118). The profile indicates only weak veering with height from 950 to 500 hPa (Figs. 3a,c). In the revision, the

above points are better described or clarified (L111, L119-121), also as suggested.

7) Except in the lowest levels, the thermodynamic profiles used seem rather dry. Is this typical for the heavy rainfall soundings during the Mei-Yu season over Taiwan?

**Reply:** As noted in the reply above (to point #6), the sounding profile used in CTL is the mean from seven profiles at 0000 UTC and thus is typical in the Mei-yu season but not necessarily conducive to heavy rainfall (L111). In the revision, rainfall information associated with the sampled days are added in Section 2.1 (L117-118) as suggested. For more moist conditions at low levels, the effects of changing near-surface RH to 92.5% and 100% are also tested in this study, and relevant results are discussed in Section 4.1 (L333-345).

8) The authors use observed SST as the lower boundary condition over the open ocean. The reviewer presumes there are spatial variations in SST in this region. How would the spatial variations in SST affect the horizontal distributions of thermodynamic fields in the mixed layer and the depth of the mixed layer? Are those being considered in the model initial conditions?

**Reply:** As described, the time-mean of NOAA analyzed SST (with spatial variations) for the period of May-June 2008 are provided at the lower boundary, coupled with a substrate model (down to a depth of 40 m). In the revision, the above configuration is better clarified (L144, L155-156) as suggested.

9) Each simulation was run for 50 hours and the first two hours are considered as the model spin up period. How is the initial spin up period determined?

**Reply:** In the revision, it is clarified that the spin-up period of 2 h is determined for the flow in the model to adjust to the topography in Section 2.4 (L180), as suggested.

10) To address the effects of thermal forcing from the land surface, it is imperative to describe the lower boundary conditions over land used in the model. The authors should also compare the simulated diurnal variations in temperature, winds and rainfall with observations very carefully. Fig. 4 shows simulated rainfall on the eastern leeside in the afternoon hours which is odd. It fails to show the effects of orographic uplift on rainfall production.

**Reply:** In the revision, the lower boundary conditions, including the SST and the substrate model (both over land and ocean) are better clarified (L144, L155-156) in Section 2, as

suggested. In Section 5, examples of our model simulations and real events are compared (please also see our reply to major point #4 above), and they agree on the convection (confirmed in satellite imageries) along the eastern slopes of the CMR, when $F_r$ is relatively small including the mixed regime (L311). The results of leeside convection are also consistent with Metzger et al. (2014), which is cited in the text in the revision (L103-104, L283, L311, 611). In addition, comparison of diurnal effects in the temperature simulations for the cases shown in Fig. 6 is added in Fig. 8 (a new figure, please also see our reply to the major comment #11 below) in the revision together with discussion (L243-245, L252-253, p.33, L763-767), along the lines as suggested.

11) Areal averaged rainfall shown in Fig. 6 is inadequate. There must be large spatial variations in rainfall throughout the diurnal cycle. What are the days used for observations (gray curve) in Fig. 6? How often do you observe southwesterly winds > 20 m s⁻¹?

**Reply:** The spatial variations of rainfall in the CTL experiment is shown in Fig. 4, while the other two model curves in Fig. 6 are shown as examples to demonstrate that large (little) diurnal variations exist under low-$F_r$ (high-$F_r$) regime. For the three cases shown in Fig. 6, a new figure (Fig. 8) is added to show the diurnal variations at the times of the peak amplitude in surface warming/cooling with discussion (L243-245, L252-253, p.33, L763-767), along the lines as suggested. In the revision, the dates used for the observed diurnal cycle in Fig. 6 are clarified both in the text and caption (L220, L747), and the observed peak strength of the southwesterly LLJ in the Mei-yu season (rarely exceeds 22.5 m s⁻¹) are also provided with references (L165-166, L520-521), both as suggested.

**3. Minor points:**

1) The figure caption for Fig. 2b is very confusion.

**Reply:** The caption of Fig. 2b is revised and clarified (L709-711), as suggested.

2) Figure 4: Should provide information on local time. The authors should also adjust the color table.

**Reply:** The information of local standard time (LST) is added in the caption of Figs. 4 and 5 (L733-734, L739-740), as suggested. The color table of Fig. 4 is also revised and updated (p.27, L729-730) as suggested.

---

## Author Comment (AC2)

NHESS-2021-196

Authors' Responses to Reviewer 2 (RC2, anonymous)

Date: 22 Nov 2021

Title: Idealized Simulations of Mei-yu Rainfall in Taiwan under Uniform Southwesterly Flow using A Cloud-Resolving Model

Authors: C.-C. Wang, P.-Y. Chuang, S.-T. Chen, D.-I. Lee, and K. Tsuboki

**1.    General comments:**

This study presents findings from idealized simulations for the island of Taiwan in which a uniform southwesterly flow is prescribed at fixed directions/speed combinations to investigate rainfall characteristics in the absence of large-scale frontal systems. In addition, near-surface relative humidity is varied and a subset of the simulations has been compared to observational data. The authors identify three rainfall regimes that correspond to different ranges of the wet Froude number and possible mechanisms for the resulting precipitation location and intensity are hypothesized. Although the paper is mostly well written and the illustrations have a good quality, it is a bit hard to see the innovation of this paper. The main result regarding the dominant process for rainfall production through mechanical uplift or thermal forcing is pretty much expected and there is a lack of evidence for their hypotheses. Although I like the general concept of idealized simulations using a real topography, I find the implementation and the connection to previous work for other islands unsuccessful so far. I would welcome a revised paper that is more physics-based, but that would probably involve substantial additional work and rewriting of the paper.

**Reply:** The constructive comments from this reviewer (Reviewer 2) are deeply appreciated, and the paper has been revised accordingly. In the revision (color-coded version), the changes made in response to Reviewer 1, Reviewer 2, and by ourselves (mostly some corrections and minor changes in English) are marked in red, blue, and orange, respectively. A point-by-point response to each of the comments from this reviewer are given below following their order. In each point, how and where the revision is made in the text is also specified.

**2.    Specific comments:**

1)    Experimental design:

●    Why do you restrict the ow direction to SW? Is the southwesterly ow during that time of the year dominant? How often are there situations without the Mei-yu front? This important

information needs to be given to determine if the model setup is representative or not.

**Reply:** As reviewed in Section 1, heavy rainfall in Taiwan during the Mei-yu season occurs predominately under the southwesterly flow regime (tropical air mass with abundant moisture). Thus, we focus on these wind directions in the present study. In the revision, the above reasoning is better conveyed to the readers more clearly (L23-24, L27, L33, L36, L51, L111, L663-664) as suggested.

- Are the fine steps of 2.5 m/s and 15 degrees really necessary? Would not a larger range of ow direction with larger steps (e.g. 5 m/s and 30 degrees) be more informative? For example, in the study of Metzger et al. (2014), the incoming wind direction for the island of Corsica has been changed in steps of 30 degrees to cover the all possible wind directions. Although Corsica is a smaller island, these previous results should be cited in this study. Furthermore, within the framework of the HyMeX project, several other publications covering island convection and terrain effects were published.

  Metzger, J., C. Barthlott, N. Kaltho (2014): Impact of upstream ow conditions on the initiation of moist convection over the island of Corsica, Atmos. Res. 145-146, 279-296, DOI:10.1016/j.atmosres.2014.04.011

**Reply:** Thank you for this opinion. Since we focus on southwesterly flow directions in the present study, finer steps of 2.5 m s$^{-1}$ and 15° are used, as described in Section 2.4. Please see our reply to the bullet point above. In the revision, the study of Metzger et al. (2014) and another relevant paper (Kirshbaum, 2011) are also cited for comparison (L103-104, L283, L285-286, L311, L586, L611-612), as suggested.

- Why do you use an integration time of 50 h? Would not 24 h be sufficient? Which day do you take for the analyses in Fig. 7 etc.? Day 1, day 2 or the mean of both?

**Reply:** As discussed in Section 2.4 and shown in Fig. 6 for a few examples, two similar diurnal cycles are produced in each run during $t$ = 2-50 h (L181-182). While the differences may be small, it is most likely still a little more representative to use the averages of two days (2-50 h), compared to the accumulation over a single day (2-26 h). In the revision, it is also clarified that the averages over days 1-2 (or 2-50 h) are shown in Fig. 7, both in the caption and at the end of Section 2.4 (L181-182, L371, L758), as suggested.

- What boundary conditions are applied in the model? Open, periodic?

**Reply:** In the revision, it is clarified that open boundary conditions are used (L161), as suggested.

- Deep convection is considered to be resolved at 2-km grid spacing, but is shallow convection still parameterized? If yes, how? Please specify.

**Reply:** No, shallow convection is also handled by the 1.5-moment bulk cold-rain scheme and not parameterized in CReSS. This is clarified in the revision (L149), as suggested.

2) I do not understand why the Froude number changes with the wind direction. If the wind speed does not change and the mountain height is constant as well, the Froude number should be independent of the ow direction unless the stability is changed. The authors should make an effort to explain how they calculate their Froude number in detail (spatial average, at what time, ...).

**Reply:** The Froude number ($F_r$, $F_r = U/Nh_0$) changes with wind direction because $U$ is the speed of wind component normal to the long axis of topography, and this is clarified in the revision (L42, L100, L189-190), as suggested. Thus, even with strong flow, the $F_r$ would still be small if the wind is parallel to an elongated topography like Taiwan (e.g., 195° in Table 3). However, if the topography is bell-shaped and does not have a long axis (as adopted in many earlier studies), the wind direction then indeed does not affect $F_r$. In several places in the text, this is also made clearer to the readers (L46-47, L100, L229, L259-260, L416-417), along the lines as suggested.

3) The authors speculate about the involved processes, i.e. terrain uplift and/or sea breeze/thermal circulations. None of these are assessed or proven in a quantitative way. Only for the CTL-run presented in Fig. 5, there is some evidence by the streamlines. I suggest to include additional material, e.g. low-level moisture convergence for establishing the impact of sea breeze on island convection.

**Reply:** For the three cases shown in Fig. 6, a new figure (Fig. 8) is added in the revision to show the diurnal variations at the times of the peak amplitude in surface warming/cooling with discussion, including that on the daytime sea breeze (L243-245, L252-253, L285, p.33, L763-767), along the lines as suggested.

4) Fig. 6: Observed precipitation starts to increase at around 20 UTC and reaches a plateau between 22-05 UTC before it further rises to the maximum value at 07 UTC. What mechanisms are responsible for the plateau?

**Reply:** We have checked the rainfall data and radar/satellite loops on those dates used to construct the observed cycle in Fig. 6 (as also better clarified in the caption). In the revision, it is explained and clarified that the plateau structure (about 0.5 mm h$^{-1}$) was mainly from migratory rainfall systems from upstream on two of the days (29 May and 4 June) during 2200-0500 UTC (0600-1300 LST), and by design, such systems are largely absent in our idealized simulations with uniform flow and no disturbances (L220-226), as suggested.

5)    L71: What are "unwanted features"? Please specify.

**Reply:** In the revision, "unwanted features" is rephrased to "undesirable features" to improve clarity of the sentence (L73), along the lines as suggested.

6)    The intercomparison to observations mostly shows a bad agreement between simulations and observational data (Fig. 10, 11, 12). Either the environmental conditions in the dates chosen do not match the model settings or other processes are missing in the model. I suggest to run realistic simulations with initial and boundary conditions from an operational model or other global analyses for these cases.

**Reply:** Indeed, some processes other than those associated with Taiwan's topography must exist (and cannot be avoided) in real conditions, such as frontal forcing, various disturbances, and low-level convergence from non-uniform flow, and even deviations from the prescribed profile and state. All these differences are not included in our idealized simulations by design. In the revision, the above points are stressed as caveats in sections 3.1 (L220-226) and 5 (L358-360, L365-369, L395-396, L428-435), along the lines as suggested. In the comparison between idealized simulations with real events of choice (when the conditions are relatively pure) in Section 5 of the previous draft was not successful, mainly because we didn't include the right data for comparison. In the revision, satellite cloud imageries at selected times are also provided (together with radar composite and the derived rainfall estimate), and they are much better to validate the model simulations (L194-195, L371, L374-378, L380-396, L402-414, L416-435, L565, L569-571, Figs. 11-13, p.36-38, L779-786, L789-791, L795-797). The reasons why both the rain-gauge data (used in previous draft) and radar composites cannot capture the convection/rainfall along the eastern slopes of the CMR are provided (L383-387, L527-529, Fig. 2b, p.24, L707-711). At various places where needed, caveats are also added or stressed in the revision to clarify possible (or likely) differences between the model results and observations (L220-226, L365-369, L381, L395-396, L408-414, L424-426, L428-435), as suggested. For heavy-rainfall cases, modeling studies using gridded analyses and full physics have been carried out, and are also cited in the revision (L425-426, L431-433), as suggested.

**3. Technical corrections:**

1) L15: local afternoon

**Reply:** Deleted as suggested (L15).

2) L17: This sentence needs to be rephrased. What is a "large angle"?

**Reply:** This sentence is broken down into two sentences to improve the readability, and it is also clarified that "large angle" means not parallel (L17-18), as suggested.

3) L40: Blumen, 1990: Blumen is the book editor for Banta (1990). Do the authors mean the Banta article here?

**Reply:** Yes, the reference is meant to be Banta (1990) here. It is now corrected in the revision (L41), as suggested.

4) L45: orographic precipitation can often be resulted → please rephrase

**Reply:** This sentence is rephrased to "… to climb over the terrain and orographic precipitation is often resulted…" to improve the readability (L46-47), as suggested.

5) L60: Wang et al., 2002, 2003: For these years, there are only entries in the references for Wang and Chen (2002, 2003).

**Reply:** Corrected to Wang and Chen (2002, 2003) here (L63), as suggested.

6) L80: $Fr \rightarrow F_r$

**Reply:** Corrected as suggested (L82).

7) L144: Murakami et al. (1990, 1994) → Murakami (1990), Murakami et al. (1994)

**Reply:** Corrected as suggested (L151).

8) L147: Sagami → Segami

**Reply:** Corrected as suggested (L155).

9)   L174: Chen and Lin (2005): Which entry is meant here, 2005a or 2005b?

**Reply:** Corrected to Chen and Lin (2005b) here (L185-186), as suggested.

10)   L184: The results of the CTL-run is …

**Reply:** Revised as suggested (L199).

11)   L185: it  behaves as designed.

**Reply:** Revised as suggested (L200).

12)   L235: regime

**Reply:** Corrected as suggested (L257).

13)   L330: …this is resulted because… → please rephrase

**Reply:** This sentence is rephrased to "… Nevertheless, with a reduced RH, the convection becomes more difficult to be triggered and thus less active at the windward side, and thus a lowered peak amount and a shift in its sub-region are resulted" to improve the readability (L353-354), as suggested.

14)   L565: Miguietta → Miglietta

**Reply:** Corrected as suggested (L613).

---

## Referee Report (RR1)

**Comments on the "Idealized Simulations of Mei-yu Rainfall in Taiwan under Uniform Southwesterly Flow using A Cloud-Resolving Model by Wang et al."**

In this study, the authors have conducted idealized simulations using a cloud-resolving model with a horizontally uniform, southwesterly flow to investigate rainfall characteristics, moist flow regimes, and the role of the complex topography in Taiwan during the Mei-yu season in the absence of Mei-Yu fronts or other weather systems. The design of idealized simulations on testing different moist flow regimes is excellent and the paper is well-written. Thus, except for the minor comments described below, I would recommend this paper be accepted with minor revision.

**Minor Comments:**

Line 59: What kind of thermodynamic effects of the topography? The authors need to clarify it.

Line 90: "The long-term climatology (1981-2010) reveals abundant Mei-yu 90 rainfall in the two-month period of May-June, with three maxima: two on the windward side of the Central Mountain Range (CMR) in southern and central Taiwan, respectively, and the third, less distinct center in northern Taiwan, roughly along the northern slope of the Snow Mountain Range (SMR)" – I would prefer to justify this sentence with the reference.

Line 118: "Shown in Figs. 3a-d, the averaged thermodynamic, moisture, and wind profiles in the vertical from these data indicate a rather uniform south-southwesterly flow (8-13 m s$^{-1}$) that veers slightly with a height from the lower to middle troposphere" This sentence is not clear (see highlighted in blue), which needs to be reworded.

Line 164-165: What about high wind with speeds more than 22.5 ms$^{-1}$? Do you have any point/explanation about if there is a high wind speed, e.g., 25, 30, 35 ms$^{-1}$.....etc.?

Line 172: Why did the authors choose these specific wind directions (210º, 240º, and 270º) and wind speeds (10, 15, and 20 m s$^{-1}$) to examine the moisture effects? Why not rest of others' direction and speeds? Are there any specific reasons? If there are any, it is better to explain here.

Lines 219 to 222: Authors mentioned that the CTL case produced poor results compared to the observation. However, the c050_210 case produced better results than the CTL case when compared with Obs. Why does c050_210 produce better results? I tend to think CTL should produce better results than other cases.

Table 1: Is there any specific reason to use Lambert conformal projection instead of Mercator projection, which is considered to be better for this region? Any reason needs to be mentioned/explained in the model and experimental part of the manuscript.

Table 3: Why are the values of moist Froude number for the case with $195^o$ direction almost constant (~0.01) for all varying wind speeds (5 to 22.5 m s$^{-1}$) cases?

Table 4: Authors found the moist Froude number for the case with $195^o$ direction almost constant (~0.01); however, the mean daily rainfall decreased; why?

Table 6: What does S, P, and M stands for needs to be mentioned in the caption.

Line 441: Do you think about the sensitivity of the terrain played on other factors? For example, what about removing the whole mountain and/or removing the mountain sequentially?

---

## Author Response (AR2)

NHESS-2021-196

Authors' Responses to Reviewer 2 (RC2, anonymous)

Date: 30 Mar 2022

Title: Idealized Simulations of Mei-yu Rainfall in Taiwan under Uniform Southwesterly Flow using A Cloud-Resolving Model

Authors: C.-C. Wang, P.-Y. Chuang, S.-T. Chen, D.-I. Lee, and K. Tsuboki

**1. General comments:**

The submitted version of the manuscript has been significantly improved over the first version. The authors have addressed all of the prior concerns to my satisfaction. I therefore believe that this article can be accepted for publication after very minor modifications which are outlined below:

**Reply:** The positive view and constructive comments from this reviewer (Reviewer 2) are deeply appreciated, and the paper has now been revised accordingly. In the revision (color-coded version), the changes made in response to Reviewer 2 and Reviewer 3 are marked in blue and green, respectively. A point-by-point response to each of the comments from this reviewer are given below following their order. In each point, how and where the revision is made in the text is also specified.

**Minor comments:**

1. p13, L412: "when the convection is relatively clean" --> what is clean convection supposed to be, do the authors mean scattered convection?

**Reply:** In the revision, this sentence is clarified to "…especially on radar and satellite images at one selected time (when the convection is less widespread)" (L414-415), along the lines as suggested.

2. Fig. 12: last plot should be labeled (h) and not (g)

**Reply:** Thank you for point this out. In the revision, the label has been corrected (Fig. 12, panel h) as suggested.

NHESS-2021-196

Authors' Responses to Reviewer 3 (RC3, anonymous)

Date: 30 Mar 2022

Title: Idealized Simulations of Mei-yu Rainfall in Taiwan under Uniform Southwesterly Flow using
    A Cloud-Resolving Model

Authors: C.-C. Wang, P.-Y. Chuang, S.-T. Chen, D.-I. Lee, and K. Tsuboki

**General comments:**

In this study, the authors have conducted idealized simulations using a cloud-resolving model with a horizontally uniform, southwesterly flow to investigate rainfall characteristics, moist flow regimes, and the role of the complex topography in Taiwan during the Mei-yu season in the absence of Mei-Yu fronts or other weather systems. The design of idealized simulations on testing different moist flow regimes is excellent and the paper is well-written. Thus, except for the minor comments described below, I would recommend this paper be accepted with minor revision.

Reply: The positive view and constructive comments from this reviewer (Reviewer 3) are deeply appreciated, and the paper has now been revised accordingly. In the revision (color-coded version), the changes made in response to Reviewer 2 and Reviewer 3 are marked in blue and green, respectively. A point-by-point response to each of the comments from this reviewer are given below following their order. In each point, how and where the revision is made in the text is also specified.

**Minor Comments:**

Line 59: What kind of thermodynamic effects of the topography? The authors need to clarify it.

**Reply:** Here, we meant the thermodynamic effects just reviewed in this paragraph, and this is clarified in the revision (L59), as suggested.

Line 90: "The long-term climatology (1981-2010) reveals abundant Mei-yu 90 rainfall in the two-month period of May-June, with three maxima: two on the windward side of the Central Mountain Range (CMR) in southern and central Taiwan, respectively, and the third, less distinct center in northern Taiwan, roughly along the northern slope of the Snow Mountain Range (SMR)" – I would prefer to justify this sentence with the reference.

**Reply:** Here, we were referring to the climatology shown in Fig. 2a, but we forgot to cite the panel explicitly. In the revision, this is clarified and Fig. 2a is cited (L90, L93), along the lines as

suggested.

Line 118: "Shown in Figs. 3a-d, the averaged thermodynamic, moisture, and wind profiles in the
vertical from these data indicate a rather uniform south-southwesterly flow (8-13 m s-1) that
veers slightly with a height from the lower to middle troposphere" This sentence is not clear
(see highlighted in blue), which needs to be reworded.

**Reply:** In the revision, it is clarified that the mean profiles here (and shown in Figs. 3a-d) are from
the seven selected soundings as just described (L118-119), as suggested.

Line 164-165: What about high wind with speeds more than 22.5 ms$^{-1}$? Do you have any
point/explanation about if there is a high wind speed, e.g., 25, 30, 35 ms-1…..etc.?

**Reply:** The observed limit in wind speed near Taiwan during the Mei-yu season is reviewed here,
and the reason to set the highest wind speed to 22.5 m s$^{-1}$ is clarified in the revision with reference to
two more studies already included in the list (L167-168), along the lines as suggested.

Line 172: Why did the authors choose these specific wind directions (210º, 240º, and 270º) and wind
speeds (10, 15, and 20 m s$^{-1}$) to examine the moisture effects? Why not rest of others' direction
and speeds? Are there any specific reasons? If there are any, it is better to explain here.

**Reply:** In the revision, it is clarified that the additional tests on moisture are a subset of those
designed to test wind direction/speed combinations, and it is so chosen without adding a large
number of extra experiments (L172-175), as suggested.

Lines 219 to 222: Authors mentioned that the CTL case produced poor results compared to the
observation. However, the c050_210 case produced better results than the CTL case when
compared with Obs. Why does c050_210 produce better results? I tend to think CTL should
produce better results than other cases.

**Reply:** In the manuscript, the reason for the difference in rainfall production in the CTL and the
observation is explained (L220-227), and in the revision, it is also pointed out that the c050_210
case, with more daytime rainfall, produces a diurnal cycle in Fig. 6 that is more similar to the
observation than the control run (L247-248), along the lines as suggested.

Table 1: Is there any specific reason to use Lambert conformal projection instead of Mercator
projection, which is considered to be better for this region? Any reason needs to be
mentioned/explained in the model and experimental part of the manuscript.

**Reply:** In the revision, it is clarified that the CReSS configuration given in Table 2 (including the projection) is similar to those adopted in previous studies with references cited (L163-164), along the lines as suggested.

Table 3: Why are the values of moist Froude number for the case with 195° direction almost constant (~0.01) for all varying wind speeds (5 to 22.5 m s$^{-1}$) cases?

**Reply:** At a fixed angle and the same stability (i.e., Brunt-Vaisala frequency $N$), the values of $F_{rw}$ would be proportional to the wind speed. At a wind direction of 195°, the values increase from about 0.01 at 5 m s$^{-1}$ (and 7.5 m s$^{-1}$) to 0.03 at 22.5 m s$^{-1}$ in Table 3, so they are small but not constant (as the value is roughly tripled like the wind speed). The values are small because the wind direction is nearly parallel to the topography (so the normal component is nearly zero), and this is already explained in the manuscript (L230-231). In Table 3, the values (for 195°) only appear to change little because they are rounded to two places below the decimal (not more places to better tell the differences).

Table 4: Authors found the moist Froude number for the case with 195° direction almost constant (~0.01); however, the mean daily rainfall decreased; why?

**Reply:** As the flow is nearly parallel to the topography, all cases with 195° direction belong to low-$F_{rw}$ regime and the relevant discussion on the reasons for the rainfall decrease with increasing wind speed is in the second paragraph of Section 3.2 (L238-248). In the revision, to better clarify, it is explicitly pointed out that all cases with 195° direction belong to this regime (L241), along the lines as suggested.

Table 6: What does S, P, and M stands for needs to be mentioned in the caption.

**Reply:** In the revision, it is clarified in the caption of Table 6 that the three letters stand for plain (P), slope (S), and mountain (M), respectively (L825), or M: mountain), as suggested. Earlier in the manuscript, the letters have been defined in the caption of Fig. 2.

Line 441: Do you think about the sensitivity of the terrain played on other factors? For example, what about removing the whole mountain and/or removing the mountain sequentially?

**Reply:** The terrain effect on other factors have been reviewed in Section 1 of the manuscript (L35-38, L39-50), including studies that employed sensitivity tests with terrain removal and/or terrain reduction (such as Wang et al., 2005). Since these studies have been reviewed, it is perhaps not

necessary (nor suitable) to mention them again in Section 6 (Conclusion and summary), as they are done with real events using a different approach (idealized simulations) as the present study.